# The World Wide Lightning Location Network (WWLLN) over Spain

Enrique A. Navarro[1,2], Jorge A. Portí[3], Alfonso Salinas[4], Sergio Toledo-Redondo[5], Jaume Segura-García[2], Aida Castilla[5], Víctor Montagud-Camps[5], Inmaculada Albert[5].

[1]IRTIC Institute, University of Valencia, Paterna, 46980, SPAIN
[2]Department of Computer Sciences, ETSE, University of Valencia, Burjassot, 46100, SPAIN
[3]Deparment of Applied Physics, University of Granada, Granada, 18071, SPAIN
[4]Deparment of Electromagnetism and Matter Physics, University of Granada, Granada, 18071, SPAIN
[5]Deparment of Electromagnetism and Electronics, University of Murcia, Murcia, 30100, SPAIN

Correspondence to: Enrique A. Navarro (enrique.navarro@uv.es)

**Abstract.** The Worldwide Lightning Location Network (WWLLN) operates a distributed network of stations which detect lightning signals at a planetary scale. Very high currents from lightning strokes radiate strong very low frequency signals in the band 6–22 kHz, detected up to 10,000 km by the WWLLN stations, which determine the time and position of the lightning
stroke detected by triangulation, similarly as global positioning systems do. Studies of the performance of the WWLLN at different areas around the word have already been reported in the literature, but similar studies for west European regions are still unavailable. This work presents a study to determine the detection efficiency and location accuracy of the WWLLN over Spain by comparing data with those of the Meteorological Spanish Agency, AEMET, during 2012, taken as a ground truth. The study provides a detection efficiency for the WWLLN of around 29 % and a location accuracy between 2 and 3 km. The
efficiency for high energy strokes is considerably higher. A study of four reduced regions with different geographical features is also considered. The peak current distribution of lightning events in these regions is obtained and the possible link to the WWLLN performances is discussed. Finally, an application of the WWLLN data for three major storms in 2020, 2021, and 2022 in the Mediterranean area of Spain demonstrates that the WWLLN is well suited for tracking the time evolution of adverse meteorological phenomena.

**Keywords:** Lightning, Geolocation, WWLLN, Global Lightning Geolocation, Terrestrial Monitoring Sensors, Atmospheric Electricity, Detection Efficiency and Location Accuracy of WWLLN in Spain.

## 1 Introduction

An important objective of regional and national lightning location networks is the detection and tracking of Cloud-to-Ground
(CG) lightning strokes. The CG lightning strokes coexist with Cloud-to-Cloud (CC) and Intra-Cloud (IC) discharges. While

the study of IC events is of great interest because they are considered the more important natural source of high frequency and very high frequency radiation (Thomas et al., 2001) and have a direct interest for air traffic controllers, for instance, the social interest in monitoring and detecting CG activity relies on the fact that CG discharges are those that mainly pose a danger to people and cause death and other economic damages, such as forest fire or other disasters. Lightning activity is also important in areas such as those of energy and telecommunications network management. In year 2023, the risk of forest fire was extremely high, due to the long period of drought affecting the Iberian Peninsula, therefore, lightning activity information may be of great social interest to prevent fire (Benito-Verdugo et al., 2023; Pérez-Invernón et al., 2023; Rodrigues et al., 2023). On the other hand, lightning discharges are closely related to storm dynamics and provide much relevant meteorological information, relevance which is currently growing, when it seems that global warming is accelerating (Ciracì et al., 2023).

Global networks, such as the Earth Networks Total Lightning Network (ENTLN), https://www.earthnetworks.com/, the Global Lightning Detection Network GLD360 of Vaisala, https://www.vaisala.com/en/products/systems/lightning/gld360, or the Worldwide Lightning Location Network (WWLLN), https://wwlln.net/, provide global Earth information for the purpose of monitoring these atmospheric phenomena and enabling users a faster location and warning of storms and other forms of severe weather hazards, such as tornadoes, downbursts, and hails, for instance. Knowing the accuracy of the data provided by these networks is very important for potential customers which are going to use or analyze those data.

Different works clearly show the interest in the use of the WWLLN as a fundamental tool to study different geophysical aspects concerning the global lightning activity in the Earth. A comparison of the lightning activity in two areas of the Congo Basin during years 2012 and 2013, based on data from the WWLLN, can be found in (Kigotsi et al., 2018). The capacity for analysis at a global scale is used in (Ccopa et al., 2021) to compare the lightning activity during years 2012 and 2013 with the Carnegie universal curve. The relation between storms, gravity waves, and their effect on the ionosphere is addressed in (Chowdhury et al., 2023) using data from the WWLLN and satellite observations from the Global Navigation Satellite System-Total Electron Content (GNSS-TEC). A local use of the WWLLN is presented in (Chowdhury et al., 2023), in which a study of the energetic electron precipitation in the Van Allen belts induced by lightning activity is carried out. New and unexpected studies naturally emerge from the existence of these global networks. In this sense, Jacobson and co-workers address the problem of identifying which part of the attenuation produced in the extremely low frequency band, from 5 to 20 Hz, is originated by reflections at the D layer of the ionosphere (Jacobson et al., 2021). It is precisely the information provided by globally-distributed stations such as those of the WWLLN what helps in designing a model to study the wave propagation in the natural electromagnetic cavity defined by the Earth's surface and the lower ionosphere.

Currently, the performances of the WWLLN are well established for different areas of the Earth, including Brazil (Lay. et al., 2004), Australia (Rodger et al., 2004, 2005), New Zealand (Rodger et al., 2006), United States (Abarca et al., 2010), China (Fan et al., 2018), and part of the Western Hemisphere (Rudlosky and Shea, 2013; Thompson et al., 2014). However, the

WWLLN technical performances at other regions are still missing. The main goal of this work is determining the performances of the WWLLN in Spain to provide accurate data for the study of lightning or related events in this zone and surrounding areas. To this aim, an initial comparative study will be carried out to determine the detection efficiency (DE) and location accuracy (LA) of the WWLLN over Spain, continental and insular regions, together with surrounding seas. The WWLLN data will be compared with independent ones, taken from the Meteorological Spanish Agency, the AEMET, which will be

considered as ground truth. These reference data from AEMET were openly accessible and cover the period from 1 January 2012 to 30 April 2012, soon after a new WWLLN station, the 69th one, was installed by the University of Valencia and which is continuously operating since June 2011. Our work will show that the WWLLN provides values for the DE and LA in the area of Spain which are higher than those reported up to the moment, with remarkable results for high peak current lightning strokes. A subsequent second study concerning four Spanish subregions with different geographical characteristics will be

addressed to detect possible variations on the WWLLN performances and their link to differences in the energy distribution of lightning strokes at these areas. It must be noted that the study presented here strictly provides the performances of the WWLLN during 2012 and that subsequent technical developments of the WWLLN make that these values must be understood of direct interest for studies focused on lightning activity evolution or as lower bounds for the current WWLLN performances in Spain and surrounding areas. Once the technical capabilities of the WWLLN in Spain are stablished, the WWLN will be

used to monitor three strong lightning and hail activity events which more recently affected the area of Valencia, at the East of Spain, in 2020, 2021, and 2022, respectively.

The paper is organized as follows. Section 2 describes the main characteristics of the WWLLN. Section 3 includes some details on the AEMET network used as reference. Section 4 shows and discusses the main results concerning the initial study of the

WWLLN performances in Spain, determining its DE and LA parameters in five cases. The whole Spanish region is firstly considered, followed by the study of four geographically different reduced regions and the possible link between the variations in the energy distribution of the storms in these areas and the WWLLN performances. A final and more qualitative example is presented in section 4.3 which shows the capabilities of the WWLLN network to successfully monitor the evolution of three severe storms which affected Valencia between 2020 and 2022. The main conclusions of the work are finally summarized in

section 5.

## 2 The WWLLN: main characteristics and present knowledge

The WWLLN under study in this work operates a ground-based planetary-distributed network of stations with Very Low Frequency (VLF) antennas, which detect lightning electromagnetic signals around the Earth. Very high currents from lightning radiate strong VLF signals in the band 6–22 kHz, which are detected up to 10,000 km. The WWLLN was deployed by the

University of Washington (USA) and the University of Otago (New Zealand), with the cooperation and maintenance of a large number of Universities and Research Institutions around the Earth. The distribution of associated active stations around the

whole Earth, slightly above active 60 stations since 2014 (Holzworth et al., 2021), makes it possible that the WWLLN achieve global location of lightning strokes at a planetary scale with a constantly improved accuracy (Dowden et al., 2002; Lay et al., 2004; Jacobson et al., 2006; Rodger et al., 2005, 2006, 2009; Shevtsov et al., 2016).


From an electromagnetic point of view, the Earth's atmosphere can be considered as an almost lossless volume located between the ground plane and the ionosphere. The system acts as a parallel plate waveguide with small losses, known as the Earth-Ionosphere Waveguide (EIWG). Lightning activity generates extremely high currents which excite electromagnetic propagating modes in the EIWG. Those modes resonating along the radial direction, i.e., between the ground and the

ionosphere, have resonance frequencies controlled by the ionosphere altitude, $h \sim 90$ km, and are known as sferic modes or, simply, sferics. For the fundamental mode, $h$ is half the wavelength, which gives a resonance frequency of around 1.67 kHz. This fundamental mode together with the first higher order modes are located in the VLF range and may propagate long distances without significant attenuation. For this reason, the WWLLN stations are designed to detect propagating electromagnetic fields in the kHz range, a band where lightning strokes excite a large amount of power and losses are low,

enabling their detection at distances around 10,000 km. This explains a successful operation of the WWLLN even when the distances between stations in the WWLLN were around 5,000–15,000 km before 2012. Finally, the detecting hardware of stations can take advantage of audio frequency systems (below 20 kHz), such as soundcards, which are cheap and easy to obtain.

When a lightning stroke happens, a sferic mode is excited and the antennas and hardware of some of the WWLLN stations around the Earth detect a time limited electromagnetic signal in the VLF band, with a duration of milliseconds. The time of arrival of the signal from the source to the antenna of each detecting station is measured using the timing signal of a satellite Global Positioning System (GPS). This time of arrival is used to calculate the distance from the station to the signal source and, finally, the location of the source is obtained by triangulation, using the distance to several stations, similarly as other

GPSs do. The detection of the sferic arrival at each station is a difficult task because of the back-ground noise. Therefore, improved trigger techniques are developed and combined with minimization methods in order to provide the correct timing of the arrival. The time of group arrival method and some improvements can be found in (Dowden et al., 2002) and in (Rodger et al., 2009). The lightning is processed and registered by the WWLLN system and is recorded as a correct detection if and only if the signal is simultaneously detected by a minimum of 5 stations. It is worth noting that care must be taken when

interpreting data of lightning location below some length scale. This is so because the distance determined by WWLLN corresponds to an equivalent VLF antenna transmitting from an effective point, but the actual lightning stroke path is not usually a vertical one, thus, the distance detected will not exactly coincide with the stroke contact point on the ground. The meaning of this distance is even more approximate because of the comparison with independent AEMET results, obtained with Low Frequency (LF) technology, where characteristic distances differ from those of WWLLN.


The WWLLN had 40 receiving sensors in 2010, providing a DE of around 11 % in 2010 for peak currents greater than 20 kA (Abreu et al., 2010) and a LA of around 5 km. A recent comparison over time of the WWLLN detection efficiency for different peak currents can be found in (Holzworth et al., 2019) for the New Zealand area. The number of active stations, the important number, increased until an almost stable number around 60 active stations since 2014, approximately. In particular, the WWLLN station at Valencia (Spain) was set in operation by the team at the University of Valencia in June 2011, being responsible of the maintenance of this station since then. This fact partially justifies the interest in assessing the still unknown performance of the WWLLN in the Spanish area, but it is not the only reason. Effectively, despite the global nature of the network would suggest that the WWLLN behavior would be similar to that known for other areas of the world, the characteristics of the Spain, with important geographical differences in relatively short distances (coasts, islands, mountain ranges, an inland plateau region surrounded by mountain regions,...), may greatly affect the storm characteristics and, therefore, the WWLLN performances in relatively short distances.

The WWLLN receiving sensors use a single 1.5 m whip antenna to measure the vertical polarization of the electric field associated to the sferic signal. The sensing procedure does not differentiate between CG and CC/IC discharges, since the whip antenna is sensitive to the vertical electric field and the two types of discharges show a similar behavior as regards this component of the electric field. The EIWG modes associated to the sferic signals are mainly excited by CG vertical strokes, although they may also be excited by strong IC and CC strokes. Therefore, differences between IC/CC and CG strokes are difficult to be inferred from the sferic signal and both IC/CC and CG are included in the signals measured by the WWLLN (Hutchins et al., 2012a).

The WWLLN data are provided to customers and members of the WWLLN in different formats. The APP files used in this paper provide the following information for each lightning stroke detected:

- Date and time in UTC. Time resolution is given in microseconds.
- Latitude and Longitude in degrees with four decimals.
- Residual fit time error in microseconds (<30 microseconds)
- Number of stations which detect the signal (a minimum of 5 stations).
- Root Mean Square (RMS) power estimation in kW from 7 to 17 kHz in 1.3 ms sample time.
- Power uncertainty (kW) in the power calculation.
- Number of stations which detect the stroke used for the power estimation. A subset of stations within the range less than 10,000 km distant from the stroke are used for the power estimation.

The whip antenna, the preamplifier, and the GPS are located outside the building on which the station is mounted. The preamplifier is wired to a soundcard. The soundcard is a typical commercial one inserted in the board of a desktop computer which has broadband connection inside the building. This computer processes the sferic time-domain signal combined with

the GPS timing signal and transmits the data to the processing stations. The antenna is anchored to the steel structure of the building to have a good ground plane and thus to provide a good signal-to-noise ratio in the sferic bandwidth (Dowden et al., 2002). The WWLLN receivers are designed to be sensitive to the vertical electric field from lightning strokes, minimizing the influence of magnetic induction, therefore, the sensors show the important property of being strongly immune to artificial VLF magnetic fields (Lay et al., 2004), signals which are difficult to isolate from industrial machines, household appliances, and other electronic systems. Minimization methods are used to obtain the time of group arrival and lightning locations. The quality of these data is given by the residual fit time error, lower than 30 μs (Dowden et al., 2002; Rodger et al., 2005, 2009).

Prior to address with this work the task of determining the features of the WWLLN in Spain, let us consider the present knowledge of this network and its performance in different areas around the world. As mentioned before, there is a reduced set of bibliography in which the WWLLN DE and LA are analyzed. The more relevant among these studies are summarized in Table 1, which includes details on the time period of the study, the area of assessment, the number of stations available in the WWLLN at that moment and the time-difference and spatial-distance criteria for considering the detected stroke coincident with a lightning stroke of the reference network. The network features in each specific area summarized in this table by means of two parameters: the DE and the LA. It is worth noting that these two quantities must be considered as total detection values, i.e., they correspond to all the detected lightning strokes, independently of their current peak amplitude. More detailed information on the DE values for specific current peak amplitudes can be found in the works referenced in Table 1. In these works, the WWLLN results are mainly compared with data from other terrestrial networks and, to a lesser extent, with satellite detection systems. These reference terrestrial networks operate continuously at a national or regional scale, while the information coming from satellites is not global and it is only available at limited periods of time, since these systems orbit over a certain area of the Earth at specific times. In addition, they mostly detect IC and CC flashes by means of photodiode detectors embedded in the satellite (Suszcynsky et al., 2000; Rudlosky and Shea, 2013; Thompson et al., 2014).

From a general point of view, the details reported in the studies in Table 1 show interesting results which must be taken into account when using data from the WWLLN. As regards the influence of the distance at which the lightning stroke happens, the work by (Rodger et al. 2006), the founders of the WWLLN, reveals a decreasing DE in the daytime for stations beyond 8,000 km and a DE being negligible for stations beyond 14,000 km. However, the DE was good between 10,000 km and 12,000 km during night time, which is a valuable information to decide the geographical distribution of stations. As regards the effect of the lightning stroke energy, low energy strokes may be dismissed mainly due to the attenuation when distances are large. Therefore, an improvement in the DE and LA for high energy strokes is expected, well above the low values included in Table 1 which, as previously mentioned, correspond to all the lightning strokes detected, independently of their peak amplitude.

Focusing on the results shown in Table 1 and references therein, they report an initial very low DE for the early WWLLN measurements in 2003, which was in the order of the one percent of the total lightning strokes detected by the reference networks, and reached values around 10 % for year 2012, the object of this study. This is a shortcoming of WWLLN compared to national or regional networks, but it must be taken into account the global scale nature of the network compared with the local or regional scale of the reference agencies. Moreover, the DE has large variations depending on the area of the Earth. Large differences are found in the DE and LA estimations shown in Table 1. The DE was assessed at 0.3 % in March 2003 in Brazil (Lay et al., 2004), while the assessment reported in Florida between April and September 2004 was about 4 % for currents larger than 50 kA in absolute value (Jacobson et al., 2006). The best data recorded by WWLLN so far was a DE of 31 % obtained in the Pacific Ocean in January 2010, a reduced area of the whole Western Hemisphere region considered in (Rudlosky and Shea., 2013) and shown in Table 1. The discrepancies in the results may be due to changes in the number and geographical distribution of active WWLLN stations, since the network has increased the number of active sensors over the years. There were eleven stations in the first evaluation in 2003, a number which was augmented to twenty stations in 2004, and around 60 active stations since 2012 (Holzworth et al., 2021). Differences seem to be also related to increasingly sophisticated processing techniques (Rodger et al., 2004, 2005, 2009). Moreover, the WWLLN has changed the distribution of active receiving sensors in different areas of the Earth. Other explanations for the discrepancies may be due to the assumed "ground truth" of the different networks used to compare with WWLLN (Abarca et al., 2010), some of them reporting a LA with errors assumed to be between 80−90 % (Lay et al., 2004; Brundell et al., 2002; Rodger et al., 2006). The technology deployment is focused in the detection of CGs, with exception of Los Alamos Sferic Array (Jacobson et al., 2006), whose DE is similar for both CG and CC/IC strokes. As regards the national and regional networks used as reference, they are devoted to the detection of both CG strokes and CC/IC strokes. In (Rodger et al. 2004, 2005) and at a regional scale, there were estimated 3.5 times more CC/IC strokes than CG ones (Mackerras et al., 1998; Soriano and de Pablo, 2007) and the WWLLN ratio of the detected CG versus CC/IC events was estimated to be roughly 2:1 (Hutchins et al., 2012b).

**Table 1.** WWLLN performance compared with other networks between 2004−2015.

| Authors | Time Period | Area | Available Stations | Criteria | DE (%) | LA (km) |
|---|---|---|---|---|---|---|
| Lay et al., 2004 | 6, 7, 14, 20, 21 Mar. 2003 | Brazil [40° W, 55° W], [15° S, 25° S] | 11 | 3 ms, 50 km | 0.3 | 20.25 ± 13.5 |
| Rodger et al., 2004 | 23, 24 Jan. 2003 | Australia [142° E,154° E], [25° S-37° S] | 11 | 3 ms, 50 km | 1.0 | 30.0 |
| Rodger et al., 2005 | 13 Jan. 2004 | Australia [142° E,154° E], [25° S-37° S] | 18 | 3 ms, 50 km | 13.0 | 3.4 |
| Rodger et al., 2006 | 1 Oct. 2003 to 31 Dec. 2004 | New Zealand [165° E, 180° E], [34° S, 49° S] | 26 | 0.5 ms | 5.4 | - |

| Jacobson et al., 2006 | 27 Apr. to 30 Sept. 2004 | Florida Circle with radius of 400 km | 19 | 1 ms, 100 km | <1.0 | 15.0 - 20.0 |
|---|---|---|---|---|---|---|
| Abreu et al., 2010 | 1 May to 31 Aug. 2008 | Canada, [41.78° N, 45.78° N], [77.48° W, 81.48° W] | 29 | 0.5 ms | 2.8 | 7.24 ± 6.24 |
| Abarca et al., 2010 | 5 Apr. 2006 to 31 Mar. 2009 | United States [25° N, 45° N], [75° W, 125° W] | 38 | 0.5 s, 20 km | 6.2 | NS: 4.03 EW: 4.98 |
| Rudlosky and Shea, 2013 | 1 Jan. 2009 to 1 Jan. 2012 | Western Hemisphere [38°N, 38° S], [165°E−17° W] | 38-66 | 330 ms, 25 km | ≤9.2 | 11.0 |
| Thompson et al., 2014 | 1 Jan. 2010 to 30 Jun. 2011 | Western Hemisphere [38°N, 38° S], [165°E−17° W] | 38-66 | 0.4s, 0.15° | ≤20 | - |
| Fan et al., 2018 | 1 Jan. 2013 to 1 Jan. 2015; | China [24° N, 40° N], [93° E, 105° E] | 70 | 0.5 s, 50 km | 10.0 | 9.97 ± 0.54 |
| Kigotsi et al., 2018 | 2005-2013 | Congo Basin [4° S, 1° N], [25° E, 30° E] [ 4° S, 1° N], [18° E, 23° E] | 11-67 | 0.5 s, 50 km | ≤7.5 | - |

As regards the effect of the lightning stroke energy, the DE of the WWLLN rises with increasing stroke peak current for both positive and negative CG lightning strokes, as it is first discussed in (Rodger et al., 2006) and later in (Fan et al., 2018). In fact, results shown in Table 1 report low values of the DE ranging from values around 1 % to 20 %, as mentioned above, partially due to the fact that this figure is a total value for all peak amplitudes. In addition to the summary data presented in Table 1, details in these studies referenced therein show that the WWLLN DE is above 50 % for CG strokes with currents greater than

40 kA, with a large variability depending on the region, providing a spatial accuracy of around 15 km. Another interesting fact also mentioned in these works is that the DE is always higher over the Oceans, although some variability is observed with the seasons (Rudlosky and Shea, 2013). The same is observed in Thomson et al. (2014), where higher values were obtained for the Pacific and Atlantic Oceans. This DE for high peak currents is good enough to resolve convective-storm cells within a larger storm complex, a large, circular, long-lived cluster of showers and thunderstorms that can cover a large region and lasts

more than 12 hours. A storm complex often emerges during the late-night and early-morning hours, it is identified by satellites and it is characterized by heavy rainfall, wind, hail, lightning, and, possibly, tornadoes (Jacobson et al., 2006).

The choice of the time- and spatial-coincidence criteria is crucial and affects the results in the DE and LA, as it can be seen in Fig. 1 of Thomson et al. (2014). It strongly depends on the characteristics of the available reference data. In this context, (Fan

et al., 2018) presents a comparison of WWLLN data with two reference measurement sets: data from national terrestrial sensors and data from satellite observations. In doing the comparison with the terrestrial network, the coincidence between lightning strokes is constrained to events happening within a time difference of 0.5 s and a distance of 50 km, however, the comparison with satellite data is not filtered for a distance of 50 km and provides better accuracy in determining the distance of the lightning

stroke. As mentioned above, the WWLLN stations are expected to detect VLF signals generated at distances of around

10,000 km. However, their effectivity worsens for low amplitude strokes, therefore, the geographical distribution of the

stations may affect the network features. This is a simplistic explanation, because there is also an influence of the propagating

conditions, land or water presence, and the noise environment in the station. However, a trend is observed in the DE in (Kigotsi

et al., 2018) by using Lightning Imaging Sensor technology, where the DE increases from around 2 % to 6 % between 2005

(23 WWLLN sensors) and 2013 (67 WWLLN sensors) in the continental areas of the Congo Basin.


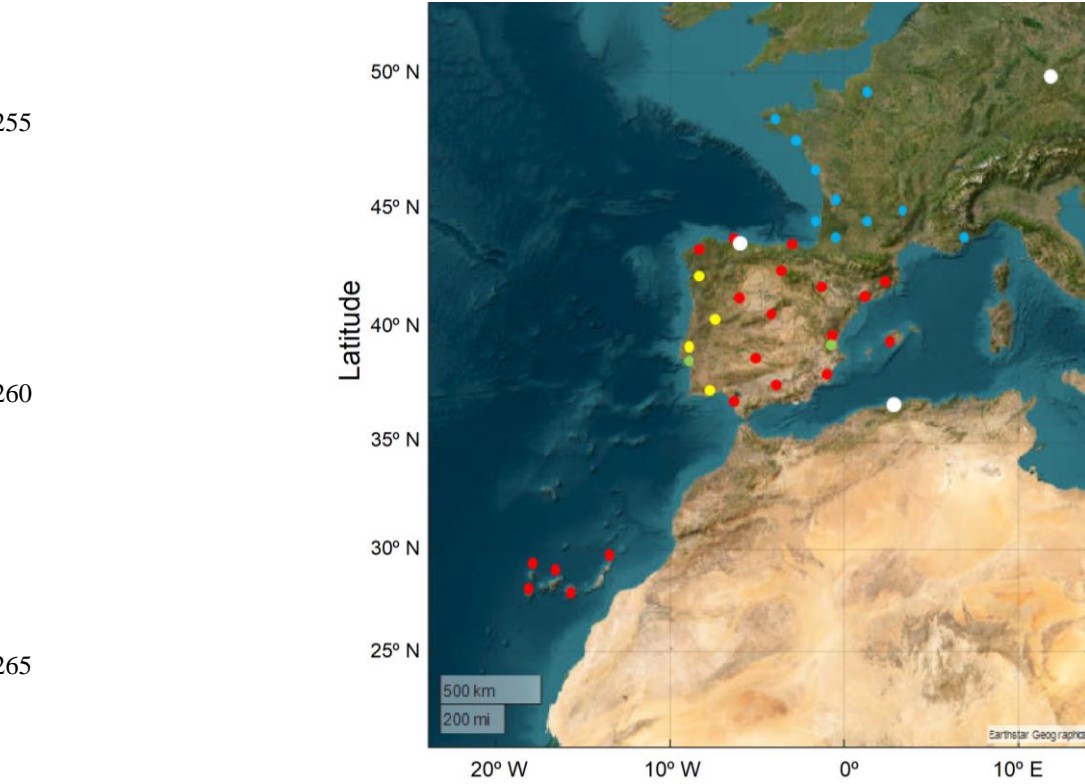

**Figure 1: WWLLN and national lightning detection networks in Spain and surrounding areas. The green color circles correspond
to the WWLLN sensors in 2012 and white circles represent the new WWLLN sensors deployed between 2020 and 2022 relative to
2012. Red circles show the positions of the AEMET sensors in 2012: 14 sensors in continental Spain, 5 in Canary Islands, 1 in
Mallorca. Four sensors of IPMA in Portugal are shown in yellow color, while blue color shows the 10 sensors of Météo-France.**

Concerning the study presented in this work, 69 WWLLN stations were operative around the world at the time period

considered, year 2012, from which around 60 are active ones. As regards the area under study, Spain, the six most relevant

WWLLN stations are shown in Fig. 1. Three stations, represented with green dots, were already operative in 2012, located in Valencia (Spain), Lisbon (Portugal), and Sheffield (England). Three new stations were deployed later, between years 2020 and 2022, shown with white dots in the figure. These new stations were deployed at Gijón (Spain), Tihany (Hungary), and Algiers (Argelia). Other stations from the National Spanish, Portuguese and French National Agencies which will be used as reference are also shown with red, yellow, and blue dots, respectively. It is worth noting that this distribution of WWLLN stations is denser than that of other areas, where stations are deployed at distances of around 10,000 km.

## 3 The reference regional lightning detection system of the Spanish National Meteorological Service, AEMET

The WWLLN performance in Spain will be determined by comparing its data with those measured by AEMET, so let us briefly describe the main features of this meteorological national agency. AEMET's sensors are spaced at distances of less than 400 km, distances well suited for the dimensions of the whole area of Spain, both the continental and the insular regions. These sensors detect the LF emissions of lightning, where LF is the International Telecommunication Union (ITU) designation for radio frequencies (RF) band, the range between 30 kHz and 300 kHz. The LF signals from lightning strokes are intense and propagate with little attenuation as surface waves over the Earth's crust. Localization of lightning strokes is carried out by AEMET stations using the IMPACT LF sensors, IMPACT ES/ESP and LS7000/7001, produced by the VAISALA company, https://www.vaisala.com/en/products/systems/lightning/single-point-sensors. This equipment is employed in around 45 national networks worldwide. The procedure starts determining the direction from which the electromagnetic signal arrives using a Magnetic Direction Finder (MDF) (Orville and Huffines, 1999; Cummins et al., 1998; Pérez-Puebla and Zanacajo-Rodríguez, 2004a, 2004b; López-Díaz et al. 2012; Cummins and Murphy, 2009). Using the information received by at least two sensors, the intersection of the lines establishes the location of the lightning. In addition, the propagation time of the signal to the sensor is determined, which depends on the distance of the sensor to the surface impact point of the discharge. Using the information from two sensors, the time of arrival delay between them determines a hyperbola with the possible locations of the discharge and the intersection of the different hyperbolas will define the possible discharge location. At least, four sensors are needed so that the location is not too ambiguous. Finally, to optimize the location, the intersection of circles is used instead of hyperbolas. Both techniques, MDF and time of arrival, are combined to obtain a better accuracy in the stroke location.

The AEMET's lightning detection network is made up of twenty electric discharge detectors distributed throughout the peninsular territory (14), the Balearic (1) and Canary archipelagos (5) (Orville and Huffines, 1999; Cummins et al., 1998; Pérez-Puebla and Zanacajo-Rodríguez, 2004a; López-Díaz et al. 2012). These detectors capture, analyze, and discriminate the electromagnetic radiation generated in atmospheric electrical discharges occurring within their range, between 50 km and 1,000 km. Through collaboration agreements, information is also received from four sensors belonging to the network of the Portuguese meteorological service (IPMA) and from sensors of Meteorage who provides data to the French meteorological

service (Météo-France) (Rodrigues et al., 2010; Santos et al., 2013). The map with these sensors is shown in Fig. 1, together with the position of WWLLN sensors installed in 2012 and 2024. These data are integrated into the system and allow optimal

coverage of the entire Iberian Peninsula and the surrounding seas.

The CG lightning detection probability of this type of network ranges between 85 % and 95 %, while its localization accuracy ranges from 100−200 m to 1 km. Likewise, the median of the peak intensity (maximum value of recorded electrical intensity) has an accuracy error of about 15−20 % and the accuracy in determining the polarity (sign of the electrical discharge) is 100 %.

As regards high-intensity lightning strokes, with intensity greater than 5 kA, a detection efficiency of more than 90 % is achieved with a LA value much lower than 0.5 km (Rodrigues et al., 2010; Santos et al., 2013).

The networks detect and keep track of lightning events, providing the time and geo-location, together with information on the originating current. The raw data files in ASCII format containing this information for each lightning stroke were available in

the web page of AEMET until the end of year 2012 (https://www.aemet.es). Currently, the availability for the research community of data from this and other national agencies is usually very difficult or expensive, what increases the interest in having more easily available data from other networks, such as the WWLLN studied here. The data used in this work as ground truth for comparison with our WWLLN data have been obtained from the AEMET web page at 2012 for the area of Spain. These open data provided the time of the lightning events with 1 s time resolution, together with information about the current

for the first lightning stroke.

The interest of national and regional agencies used as reference is directed to the detection of CG lightning, disregarding the detection of IC/CC strokes. The IC/CC lightning strokes were initially registered by the stations, however, most of them were discarded at post-processing by taking into account their calculated current. Some IC/CC lightning strokes could not be filtered

out because of their strong current, as it happens in the USA National Lightning Detection Network (NLDN) and the Canadian National Lightning Detection network (CLDN) (Abarca et al., 2010; Fleenor et al., 2009). Thus, an unknown reduced percentage of IC/CC is included in the files. There is not a clear current threshold to differentiate CG from IC/CC, which explains the small percentage of IC in the data. Some proposals have been made to distinguish between IC/CC or CG by analyzing the rate of change of the electromagnetic field, which can be obtained from the time-domain measurement of the

signal (Rakov and Uman, 2003). The misclassification IC/CC-CG was first addressed in 1994/1995 by the USA NLDN (Cummings et al. 1998; Wacker and Orville 1999; Jerauld et al. 2005; Orville et al. 2002; Cummings et al. 2006; Biagi et al. 2007). Typically, low current IC/CCs are erroneously classified as CGs, and some proposals were made to discard positive CG strokes with peak currents less than 10 kA, or reclassify them as IC (Grant et al., 2012).

## 4 The WWLLN performance in the area of Spain

A study concerning the features of the WWLLN in different areas of Spain is presented and discussed in this section, followed
by an example of application of the WWLLN stations. First, section 4.1 shows the network performances for the whole Spanish
area during 2012, soon after the Valencia station was deployed by two authors of this work. Storms are strongly affected by
geographical features; thus, it seems reasonable to think that the WWLLN performances may also be affected by these
geographical differences. In this sense, section 4.2 presents a second similar study concerning four qualitatively different

Spanish subregions to find possible inhomogeneities in DE values at the Spanish area. The four regions comprise an inland
region, together with three different regions including mostly sea areas or land-sea transitions. Once the quantitative studies to
determine the WWLLN DE for whole area of Spain and these four subregions are presented, a final qualitative application of
the WWLLN to monitor three severe storms at Valencia, at the East coast of Spain, in 2020, 2021, and 2022, is presented in
section 4.3.


Fig. 2 describes the different areas of these studies. Red and orange for the first study presented in section 4.1 The green, cyan,
dark blue, and magenta regions correspond to the areas considered in section 4.2 which will be described in more detail there.
The cyan region also approximately corresponds to the final monitoring application presented in section 4.3

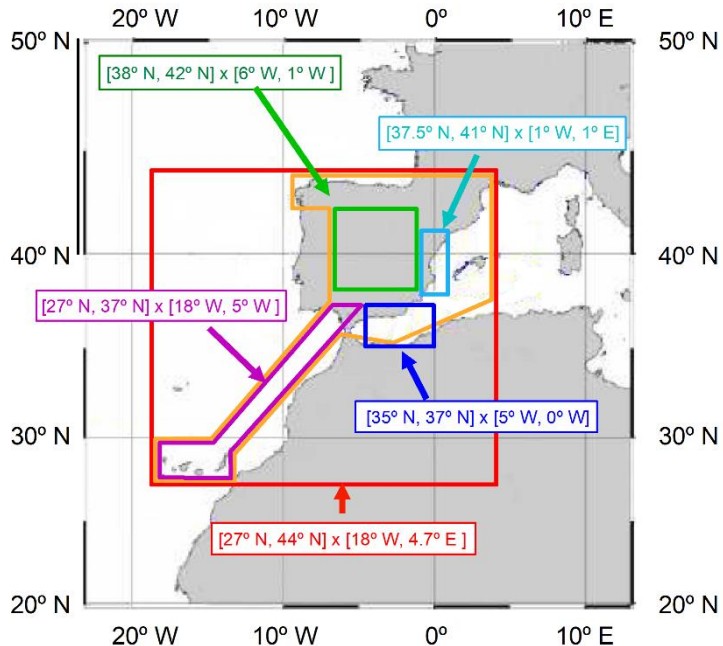


**Figure 2: Different areas for the studies presented in this work.**

## 4.1 Detection Efficiency and Location Accuracy of the WWLLN in Spain

The WWLLN performance in the area of Spain is first analyzed using data from this network (http://www.wwlln.com) during the period from 1 January 2012 to 30 April 2012. The data were generated using the most recent technique described in the work of Rodger et al. (2009). Two reasons support the choice of this time span. First, data from AEMET were openly available to the authors at that period, and second, this period was close in time to the moment at which the station in Valencia (Spain) was deployed by the team of this work in the year 2011. It must be noted that the objective of this paper is not comparing AEMET and WWLLN in order to determine which is the best network, since they have different local and global objectives, respectively. The main goal is to determine the DE of the WWLLN in the whole region of Spain, which includes continental and insular regions, using the AEMET data as true data (Abarca et al., 2010). This study is the first one analyzing the WWLLN performances at Western Europe, specifically at the Spain, where large geographical differences happen at relatively low distances. The use of data from 2012 does not invalidate the conclusions of the first study, although, bearing in mind the technical improvements of both WWLLN and AEMET since 2012, the results presented here must be considered as a lower bound for the current network performances, which, most likely, will have been improved at present days.

The time span of the lightning data chosen is usually a period with low seasonal activity in the Iberian Peninsula. The main storm activity in Iberia prior to 2012 was typically distributed in the period May−September, in which around 84 % of the storm phenomena with lightning events were detected (Soriano et al., 2005). Although the period under study had a low activity in terms of lightning strokes, the AEMET data still contain a significant number of 20,651 lightning strokes in the whole Spanish region. The 2012 AEMET data file has 20,651 lines, each line has 9 columns with the following data: month, day, hour, minute, second, discharges, peak current, latitude, and longitude.

To analyze the WWLLN detection efficiency relative to the AEMET network, we look for time and location coincidences within a given deviation to identify the strokes events shared by both networks. Several criteria have been used by different authors to define shared lightning strokes and, thus, to establish a coincidence. The particular criterion chosen greatly depends on the characteristics of the available data of the independent networks. These reference networks are assumed to provide true data, since their certainty is reported to be above 90 %. Obviously, the coincidences in time and location are used with a given tolerance or deviation from the ground truth. Lay et al. (2004) and Rodger et al. (2005) used both a time deviation of 3 ms and a space deviation of 50 km to establish the coincidence. Jacobson et al. (2006) used a time gap within 1 ms and a maximum distance of 100 km. When the data available have a high temporal resolution, the time criterion alone seems good enough to establish shared events, i.e., there is no need of using a combined spatial coincidence. This explains why the work by Rodger et al. (2006) for New Zealand and Abreu et al. (2010) for the area of Toronto only impose a time difference of 0.5 ms to decide coincident events.

As regards the study in the area of Spain presented with this work, the lightning activity data available from the AEMET have a time resolution of 1 s. This coarse time resolution forces the use of both temporal and spatial coincidence criteria to ensure confidence in this analysis. As the AEMET data were given with a resolution of 1 s, we establish a maximum time difference of 0.5 s between AEMET and WWLLN strokes to define the temporal coincidence. This large time tolerance is far from the range 0.5–3 ms used by other researchers (Rodger et al. 2005, 2006; Jacobson et al. 2006), however it is the same as in (Abreu et al.; 2010; Abarca et al., 2010), and the more recent work by Fan et al. (2018). This time coincidence is combined with a spatial coincidence of 20 km. Therefore, a WWLLN lightning stroke is shared by the AEMET reference if both events happen with a difference in time lower than 0.5 s and the distance between them is below 20 km.

In this first assessment study, the region of the world under consideration is the whole area of Spain and small near areas of the Atlantic Ocean and the Mediterranean Sea, inside the area defined by the latitude interval [27.39° N, 43.83° N] and the longitude interval [18.01° W, 4.66° E] (red rectangle in Fig. 2). The limits of this rectangular area are defined by the maximum and minimum latitudes and longitudes of the AEMET available data, which are spatially filtered to reduce to the non-rectangular orange region in Fig. 2, exclusively describing the Spanish area. As regards the WWLNN network, it collects global Earth data, therefore, the very large files contain all the registered events and these data must be geographically filtered to obtain data for this same area.

Figure 3a depicts the lightning strokes detected by AEMET in the Spanish orange region of Fig. 2 during the period January 2012 to 30 April, 2012, located as green dots in the map. The green dots in Fig. 3a comprise a total of 20,651 lightning strokes detected by AEMET, which serve as reference for the WWLLN data during the above-mentioned time period. We look for the AEMET lightning strokes that are coincident with WWLLN data following the above-mentioned criteria of time and spatial coincidence. In the WWLLN, lightning stroke data around the whole Earth are stored in files on a day-to-day basis. Each file corresponds to one day and has a size of about 30 Mb. Therefore, we first obtain the WWLLN files for the period 1 January 2012 to 30 April 2012, which occupy a total 3.96 Gb. Later on, we filter the data of these files to extract the data inside the area defined by [27.39° N, 43.83° N] x [18.01° W, 4.66° E] (red area in Fig. 2). This data file is a smaller one, with a size of 2.7Mb and includes 54,079 lightning strokes. This last file was used for further processing in order to compare it with the AEMET data. Although it does not correspond to the same geographical area yet, it has a manageable size. Finally, this file was spatially filtered with the AEMET spatial filter (orange region of Fig. 2), providing a small file having the WWLLN data reduced to the Spanish region, with a total of 12,855 lightning strokes.

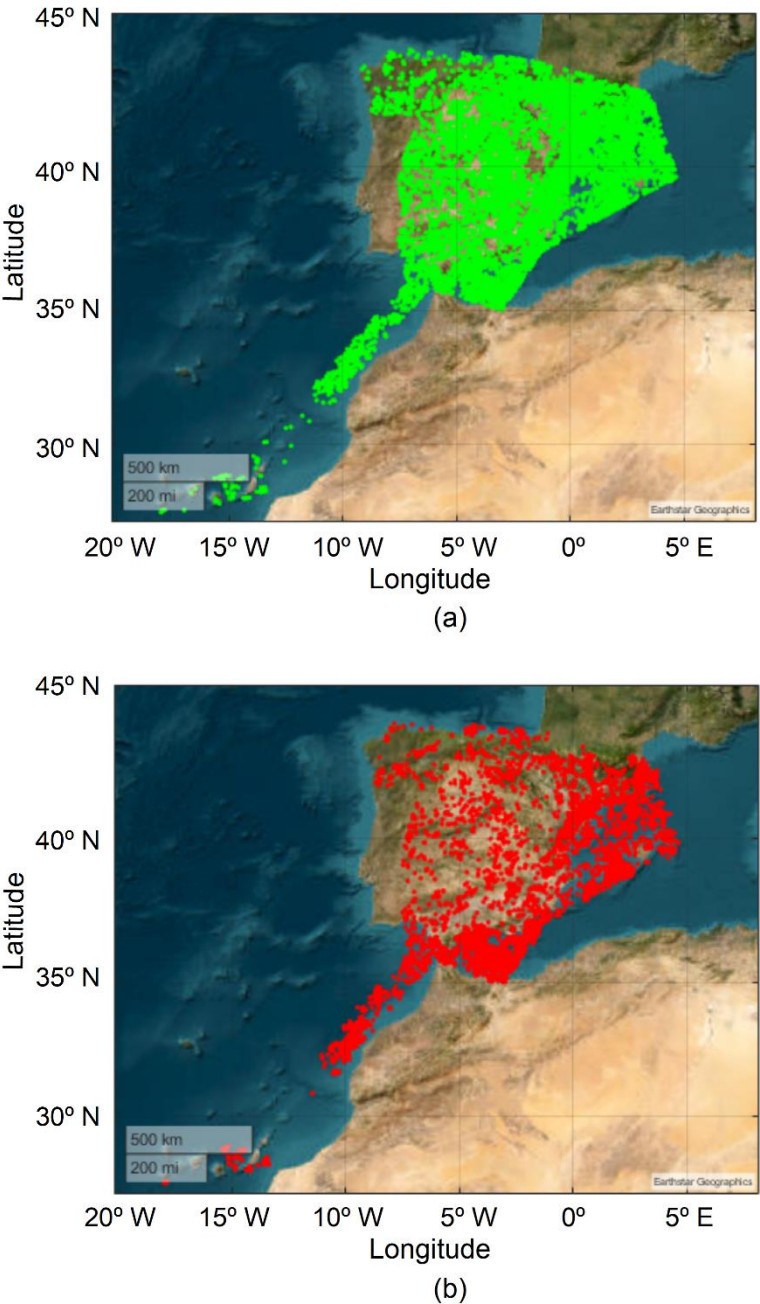

**Figure 3: (a) Lightning strokes detected by the AEMET network during the period 1 January 2012 − 30 April 2012. (b) Strokes detected simultaneously by the WWLLN and the AEMET networks during the period 1 January 2012 − 30 April 2012.**

To identify the number of lightning strokes detected by AEMET which are also detected by the WWLLN and thus determine
the DE of the global network, a correspondence is searched for in the file of the WWLLN data for each lightning stroke in the
file of AEMET data. In the first stage, we look for temporal coincidence of each AEMET stroke with WWLLN strokes using
a maximum time deviation of 0.5 s. In the second step, the maximum distance of 20 km between the AEMET and WWLLN
strokes is checked. In doing so, we obtain that 5,904 out of 20,651 AEMET lightning strokes match with one of the 12,855
lightning strokes detected by WWLLN. These coincidences are plotted in Fig. 3b and are used for further analysis and to
determine the DE and the LA. These coincidences yield a WWLLN DE of 29 %, relative to the AEMET network. The rest of
lightning detections of the WWLLN can be considered true detections without any comparison frame, most likely because
they are CC/IC lightnings, which can be detected by the WWLLN but not by AEMET. This DE of 29 % for the WWLLN in
the whole region of Spain is a considerably good result when compared with those included in Table 1. Although, in principle,
it may seem that the high density of stations in Spain may cause this relatively high DE value, this is probably not the case
since close lightning strokes usually saturate the nearest receivers. The explanation is probably more linked to the higher DE
values usually obtained in sea areas and which has already reported in the literature (Rudlosky and Shea., 2013; Thomson et
al., 2014). The geographical peninsular characteristics of Spain, with an important part surrounded by the Mediterranean Sea
and the Atlantic Ocean, seems to indicate that the effect of the large coastal regions prevail over the effect of more reduced
inland regions and may justify this relatively good DE value. This aspect will be tested in section 4.2 by considering
geographically different subregions and energy distribution of lightning strokes.

The distribution of the location error for the detected lightning strokes, according to above criteria, are presented in Fig. 4. The
location error has a maximum probability at 3.5 km for the interval 0−20 km. The distribution of locating errors along longitude
($\Delta x$) and latitude ($\Delta y$) are shown in Fig. 5 and as a scatter plot in Fig. 6, in which a slightly systematic error in the location is
observed in northward and westward directions. The standard error is larger in the West-East direction, since the Gaussian
broadens in this direction.

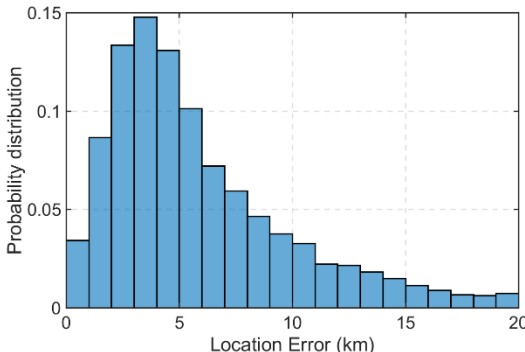

**Figure 4: Location error for the lightning strokes and probability distribution for the WWLLN correctly detected according to the criteria 0.5 s−20 km.**

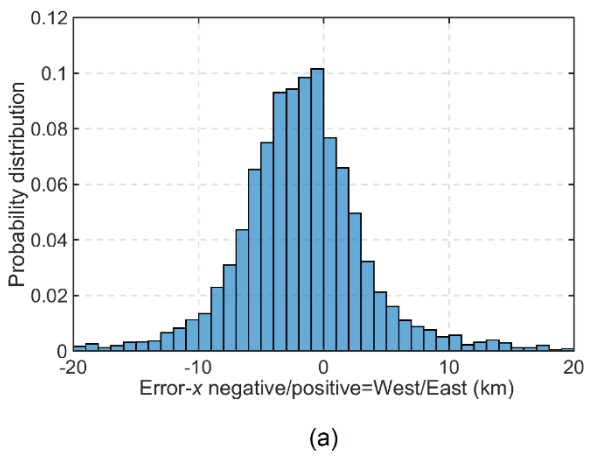
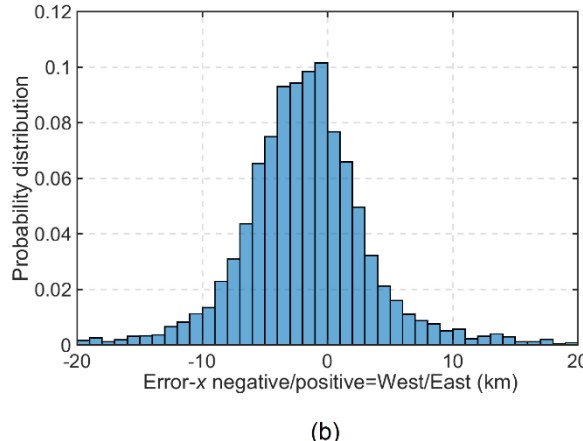

(a)

(b)

**Figure 5: Location error along longitude and latitude: (a) Error in km along longitude. (b) Error in km along latitude.**

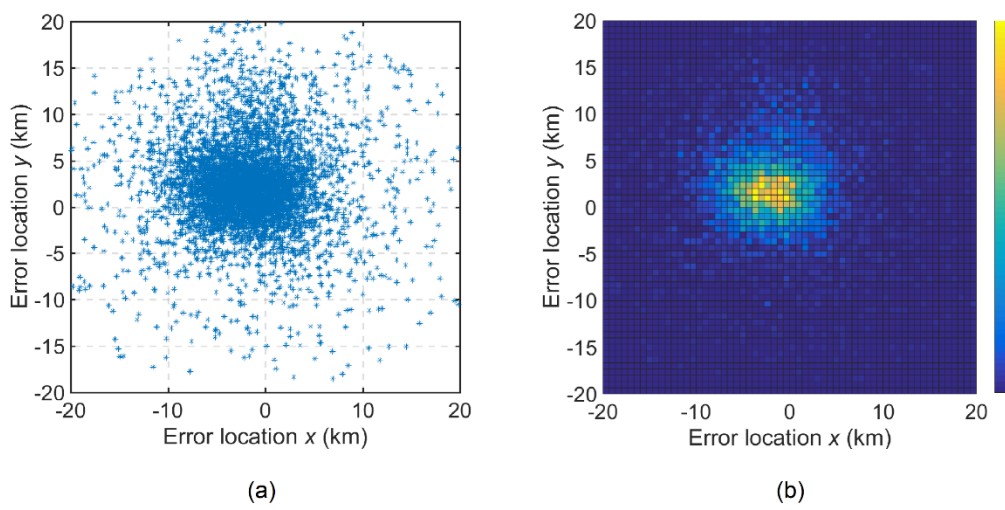

(a)

(b)

**Figure 6: Location error for the lightning strokes along *x*-longitude and *y*-latitude, at region [27° N−44° N] x [18° W−4.7° E]. (a) Error in km. (b) Color code showing the number of lightning strokes at each *x-y* error.**


Quantitative data concerning the LA for the WWLLN at Spain can be inferred from the Gaussian shaped probability distributions depicted in Fig. 6. More specifically, the average error along the West-East direction is $\Delta x$=-1.8 km (minus indicates a deviation towards west), 95 % confidence interval [-1.9, -1.6] km, while the average error along the South-North direction is $\Delta y$=2.2 km, 95 % confidence interval [2.1, 2.3] km. The deviation is larger along West-East ($\sigma_x$ =5.3 km) than along South-North directions ($\sigma_y$ =4.8 km).

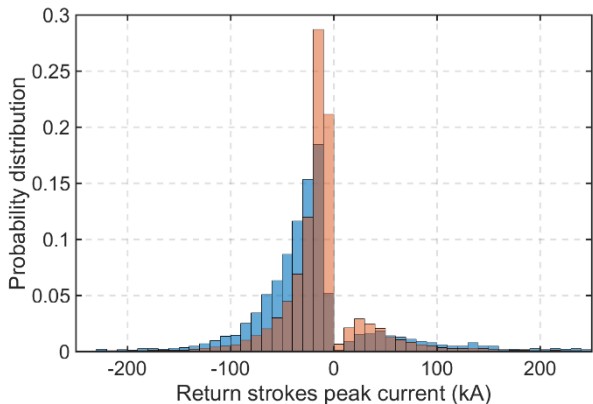

**Figure 7: Distribution of AEMET return strokes detected by the WWLLN, in blue color, and total AEMET return strokes, in orange color.**


The distribution of AEMET strokes in bins of 10 kA, together with the distribution of strokes also detected by the WWLLN, is shown in Fig. 7. The distribution of peak currents is shifted towards negative strokes, which is in good agreement with references of Table 1. There are 3,471 CG positive lightning strokes (positive peak current), versus 17,180 negative lightning strokes in the AEMET data, a 16.8 % of the detected CG strokes. As regards the WWLLN, the peak current is assigned as the

one corresponding to the matched AEMET stroke. The same ratio between positive and negative lightning strokes is preserved for the subset of 5,904 AEMET strokes also detected by the WWLLN. The figure shows the usual distribution of negative and positive CG strokes: nearly 90 % of the global lightning activity corresponds to negative peak currents. The average peak current in negative CG is -25.40 kA and in positive CG is 8.59 kA for the AEMET data. As regards the lightning strokes also detected by WWLLN, the average results are -37.9 kA for negative CG strokes and 14,0 kA for positive ones. These results

show a clear shift of the WWLLN operation towards detection of high-energy lightning strokes.

To establish the dependence of the network features on the energy of the lightning strokes, the distribution of the DE is calculated in bins of 2 kA. The result is mapped in Fig. 8, where each point (red circles) represents the DE for an interval of 2 kA. These discrete results are smoothed with a five-point mobile average (line in blue color) and standard errors in bars are

also included. The information provided by both data in Fig. 8 shows that the DE increases with the peak current, however, the DE looks slightly noisy for both positive and negative high energies above 100 kA, which is likely due to the small amount of available data (see Fig. 7 for peak currents greater than 100 kA in absolute value). Despite these slight fluctuations observed for high energy strokes, Fig. 8 shows that the DE of the WWLLN is remarkably good for lightning strokes with high peak currents, the more dangerous ones. Results of Fig. 8 are very similar to previous works referred to in Table 1 (Abarca et al.,

2010; Rodger et al., 2006; Fan et al., 2018).

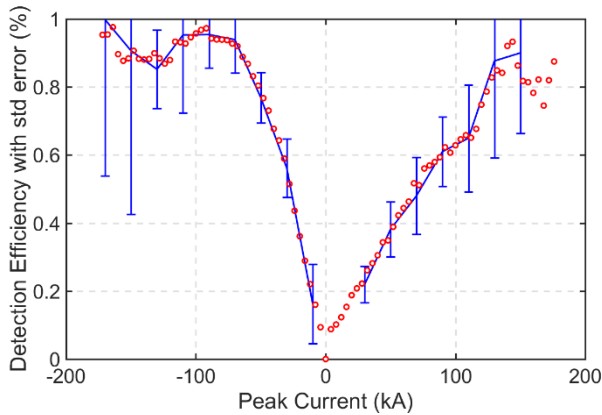

**Figure 8: Detection efficiency of the WWLLN versus lightning stroke energy. Data correspond to a bin size of 2 kA. Data smoothed are shown with the blue line.**


## 4.2 Detection efficiency and location accuracy in four reduced areas of Spain.

In order to address the possible effects of geographically features on the DE and the LA, we now restrict the analysis to four particular regions with different geographical characteristics. Firstly, the WWLLN performances are studied for a reduced inland and geographically-uniform region, the Plateau Spanish region. Three sea and land-sea regions are considered next: the

East Spanish Mediterranean coast, the South Spanish Mediterranean coast and the West African Atlantic coast. The first sea region includes a transition between a coastal and a maritime area near Valencia in which severe storms usually happen at the beginning of autumn. The second includes the Alboran Sea, which is directly affected by strong marine currents originated at the Straits of Gibraltar. Finally, the West African Atlantic coast is mostly a maritime region affected by ocean currents. The differences in the geographical characteristics in these regions produce important differences for the climate at these areas.

The aim of this subsection is to determine if these differences are reflected in the DE and the LA of the WWLLN.

The first region considered is defined by latitude [38° N, 42° N] x longitude [6° W, 1° W], which is inside the plateau area of the Iberian Peninsula (green rectangle in Fig. 2). Geographically, it is a homogenous region which avoids the main mountainous regions which surround it. In this case, the AEMET file contains 3,389 lightning strokes, while the WWLLN file has 1,229

strokes. A total of 435 of the lightning strokes detected by AEMET are also detected by WWLLN, which means that the DE for the Spanish Plateau is 13 %.

Figure 9 shows the errors along latitude and longitude for this reduced area. The results in this figure yield an average location error along the West-East direction of $\Delta x$=-2.4 km, the confidence interval of 95 % is [-2.9, -1.9] km, and standard deviation

$\sigma_x$=5.3 km. The average location error along the South-North direction is $\Delta y$=1.3 km, the confidence interval of 95 % is now

[1.0, 1.7] km, with standard deviation $\sigma_y$=3.9 km. These results for $\Delta x$ and $\Delta y$ are not significantly different from the results obtained using the larger area of section 4.1. However, the results for the scattered plot of $\Delta x$, $\Delta y$ seem better than previous ones, with lower standard deviation for the South-North direction, as it can be seen by the differences between Figs. 6 and 9. Effectively, Fig. 9 shows data more horizontally concentrated around their mean value than those represented in Fig. 6, which

corresponds to lower standard deviations and, therefore, better location error. For this area and the AEMET data, the average negative peak current in negative CG strokes is -17.92 kA and 10.26 kA in positive CG ones. As regards the strokes matched by the WWLLN, the corresponding average results are -26.4 kA for negative CG strokes and 29.4 kA for positive CG strokes. In this area, the differences in absolute value between positive and negative CG lightning strokes are lower than in the former larger area. This is probably related to the different characteristics of the storms and more likely due to the influence of

geographical features than to the characteristics of the sensors.

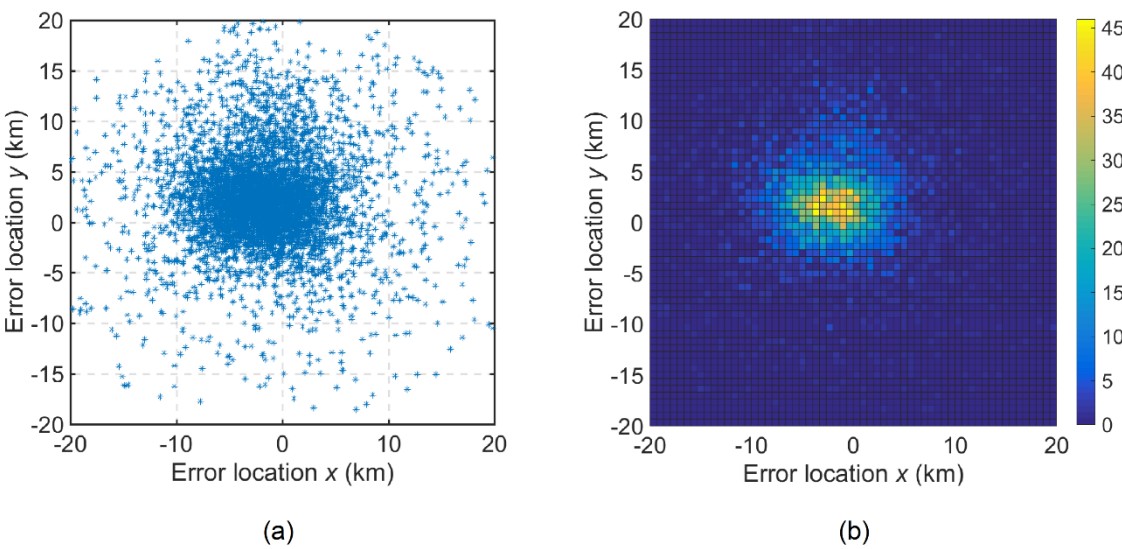

(a)                                         (b)

**Figure 9: Location error for the lightning strokes along *x*-longitude and *y*-latitude for the plateau region, [27° N, 44° N] x [18° W, 4.7° E]. (a) Error in km; (b) Color code showing the number of lightning strokes at each *x-y* error.**


The second subregion considered here includes coastal and maritime parts. The region under study correspond to a small area of the West Spanish Mediterranean coast, longitude and latitude limits given by [37.5° N, 41° N] x [1° W, 1° E], which is plotted with cyan color in Fig. 2. The comparative study of the lightning activity detected by the WWLLN with the AEMET data provides similar results as those shown in Fig. 9 (not represented). A total of 2,495 lightning strokes are detected by

AEMET for this area, of which 558 ones were also detected by WWLLN, giving a DE of 22 %. As regards the location error, the results yield an average location error along the West-East direction of $\Delta x$= -1.8 km, the confidence interval of 95 % is

given by [-2.1, -1.5] km, with a standard deviation of $\sigma_x = 5.3$ km. The average location error along the South-North direction is $\Delta y$=2.1 km, the confidence interval of 95 % is [1.9, 2.4] km, standard deviation $\sigma_y = 4.4$ km.

The third subregion, the South Spanish Mediterranean coast, is determined by longitude and latitude limits [35° N, 37° N] x [5° W, 0° W]. This region is plotted with dark blue color in Fig. 2. This area also includes a transition between land zones and the Mediterranean coast, but now the area is close to Straits of Gibraltar, with strong marine currents not present in the previous region considered. A total of 4,104 lightning strokes are detected by AEMET for this area, 2,179 of them are also detected by WWLLN, which yields a DE of approximately 53 % for this area. The location error along the West-East direction is

$\Delta x$= -1.8 km, with confidence interval of 95 % given by [-2.0, -1.7] km and standard deviation $\sigma_x = 5.3$ km. Similarly, the average location error along the South-North direction is $\Delta y$=2.1 km, the confidence interval of 95 % is [1.9, 2.4] km and the standard deviation is $\sigma_y = 5.7$ km.

Finally, the fourth reduced region considered, the West African Atlantic coast, is delimited by longitude and latitude given by

[27° N, 37° N] x [18° W, 5°W]. It is mostly a maritime area which includes the Canary Islands and the Atlantic Ocean zone at the West African coast. This region is depicted with magenta color in Fig 2. A total of 1,247 lightning strokes were detected by AEMET for this area, 613 of them also detected by WWLLN, which yields a DE value around 49 %. The location error for this area along the West-East direction is $\Delta x$=-1.4 km, the confidence interval of 95 % is [-1.8, -1.1] km and the standard deviation obtained is $\sigma_x = 5.8$ km. Along the South-North direction, the location error is $\Delta y$=1.7 km, the confidence interval of

95 % is [1.3, 2.0] km and the standard deviation is $\sigma_y = 5.3$ km.

**Table 2: Location accuracy for West-East and South-North directions and Detection Efficiency for the WWLLN in the studies of subsections 4.1 and 4.2. Comparison is made with AEMET reference data from 2012.**


| Region | West-East, $\Delta x$ (km) | 95 % CI | South-North $\Delta y$ (km) | 95 % CI | DE (%) |
|---|---|---|---|---|---|
| Spain (orange in Fig.2) | -1.8 | [-1.9, -1.6] | 2.2 | [2.1, 2.3] | 29 |
| Spanish plateau (green in Fig. 2) | -2.3 | [-2.8, -1.8] | 1.4 | .[1.0, 1.7] | 13 |
| East Spanish Mediterranean coast (cyan in Fig. 2) | -1.8 | [-2.1, -1.5] | 2.1 | [1.9, 2.4] | 22 |
| West African Atlantic coast (magenta in Fig. 2) | -1.4 | [-1.8, -1.1] | 1.7 | [1.3, 2.0] | 49 |
| South Spanish Mediterranean coast (dark blue in Fig. 2) | -1.8 | [-2.0, -1.7] | 2.7 | [2.5, 2.9] | 53 |

Table 2 summarizes the efficiency and location error for Spain and the four subregions studied in sections 4.1 and 4.2 according to AEMET reference data from 2012. Consistent $\Delta x$, $\Delta y$, are observed for the location accuracy. As regards the DE, the comparison of the reduced areas points to a higher value of DE for areas containing sea zones, specially higher at those sea areas were strong maritime currents are more relevant.

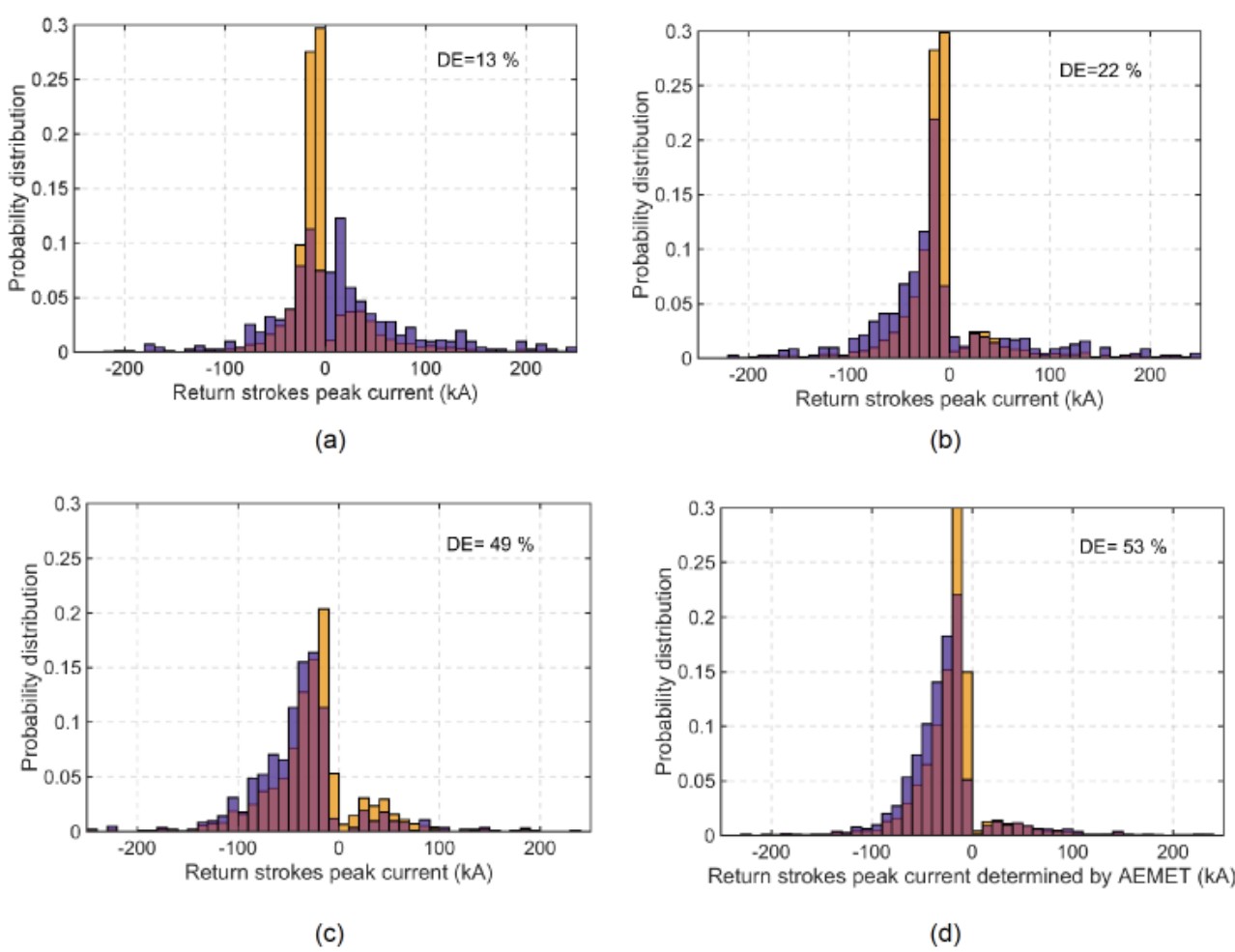

Figure 10: Distribution of AEMET return strokes also detected by the WWLLN, in blue color, and total AEMET return strokes, in orange color for different subregions: a) Spanish Plateau, b) East Spanish Mediterranean coast, c) West African Atlantic coast, and d) South Spanish Mediterranean coast.

As previously mentioned, the high value of DE at sea regions was already reported in the literature (Rudlosky and Shea., 2013; Thomson et al., 2014) and it could be related to a difference in the energy distribution of lightning strokes towards in sea areas. To support this statement, the peak current distribution of the lightning strokes detected for each subregion has been calculated. The results are shown in Fig. 10. Effectively, it can be appreciated from this figure that the continental area at the Spanish Plateau presents an important distribution of lightning strokes at low energies. On the contrary, the presence of high energy

strokes increases in the other three areas containing land-sea transitions in the following order: East Spanish Mediterranean coast, West African Atlantic coast, and South Spanish Mediterranean coast. This, combined with the results shown in Fig. 8 describing the increase in DE with current peak, seems to explain the differences in the DE obtained for the different subregions considered and draws attention on the variability of this efficiency if important geographical differences are present. In this sense and as regards the high value for the DE obtained for WWLLNN in the whole region of Spain compared to that in other

regions included in Table 1, it seems that its peninsular geographical characteristics, with a large contribution from its coasts compared to that of the inland Spanish areas, may explain this relatively high DE value of 29 %.

## 4.3 Three severe meteorological events at the Spanish Mediterranean coast.

Once the technical features of the WWLLN are determined, the network data can be useful in different applications. The

following is an example of the use of the WWLLN to monitor the evolution of three major lightning and hail storms that affected the Valencia region the following days: 18 April 2020, 30 August 2021, and 17 August 2022. The region under study is the small area of the Spanish Mediterranean coast considered in the last study of the previous subsection, approximately corresponding to the area plotted with cyan color in Fig.2.

Figures 11 to 13 show the results for the three storm events. Figures 11a, 12a, and 13a are screenshots taken directly from AEMET website, https://www.aemet.es/, while Figs. 11b, 12b, and 13b have been generated with WWLLN data. In these figures, the locations of the lightning discharges are indicated with dots. The color code is used to temporally locate the lightning discharges in one-hour bands. For each day and for each one-hour interval starting from 00:00 h to 23:00 h, the lightning location is plotted with a different color, which allows visualizing each storm time evolution.

The storm of 18 April 2020 is shown in Fig. 11. The lightning strokes detected by the AEMET are shown in Fig. 11a, while Fig. 11b shows the strokes detected by the WWLLN. According to the AEMET data, there were more than 11,000 lightning strokes in Spain land and sea, and 510 CG lightning strokes in Valencia area. The storm of 30 August 2021 is shown in Fig. 12. Again, Fig. 12a shows the lightning strokes detected by AEMET, while Fig. 12b shows the events detected by the WWLLN.

Finally, similar plots for the storm of 17 August 2022 are shown in Fig. 13a for AEMET and Fig. 13b for the WWLLN data, respectively. There were 28,666 lightning strokes in Spain during that storm; in Valencia region there were 810 CG.

.

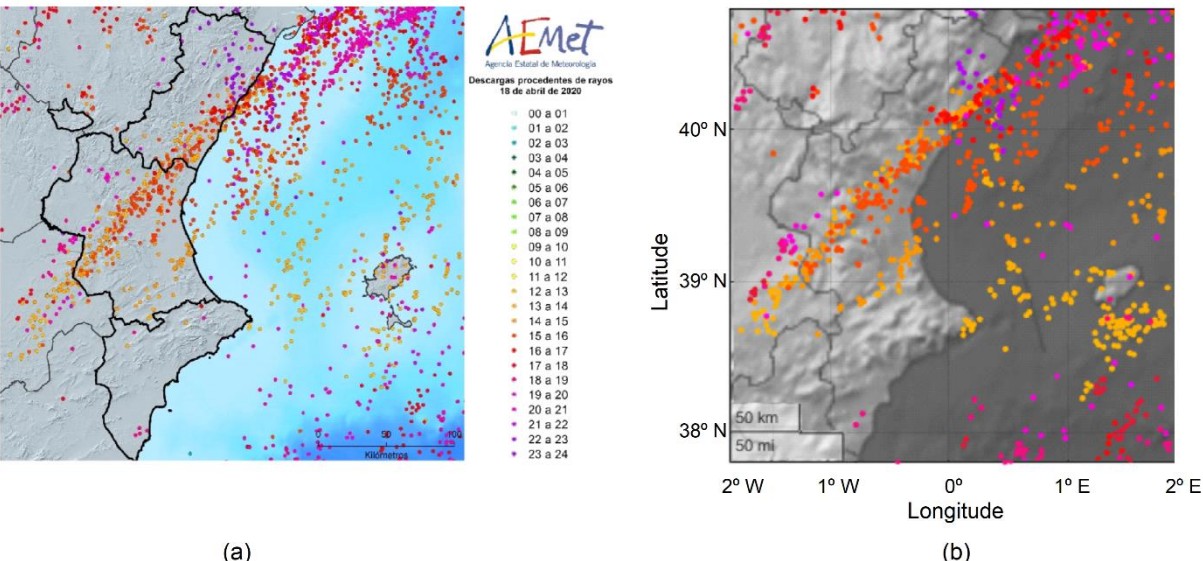

(a)                                                                    (b)

**Figure 11: The lightning storm of 18 April 2020 on the Spanish Mediterranean coast. Location of the lightning strokes with circular dots, different colors for different time periods: (a) AEMET. Image source: AEMET_C.Valenciana@AEMET_CValencia. 2020. Accessed via https://twitter.com/AEMET_CValencia/status/1251787766284857344/photo/2 ; (b) WWLLN.**

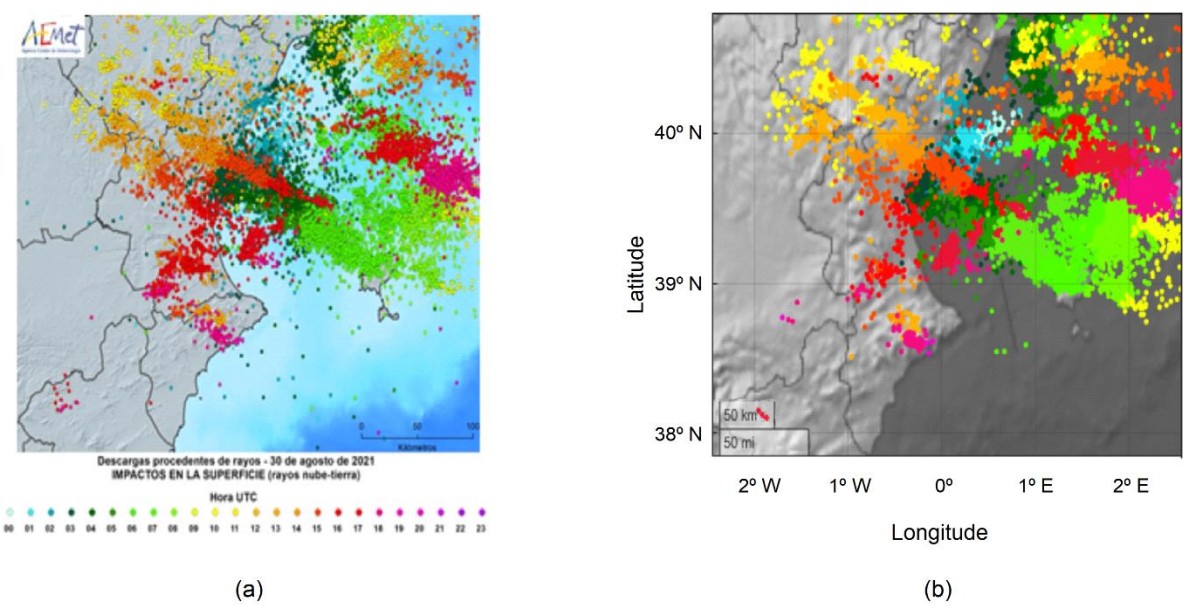

(a)                                                                    (b)

**Figure 12: The lightning storm of 30 August 2021 on the Spanish Mediterranean coast. Location of the lightning strokes with circular dots, different colors for different time periods: (a) AEMET. Image source: AEMET_C. Valenciana@AEMET_CValencia. 2020. Accessed via https://x.com/AEMET_CValencia/status/1432604874206846978 ; (b) WWLLN.**

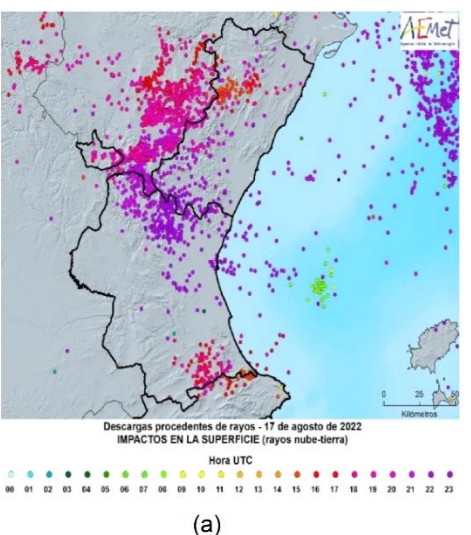
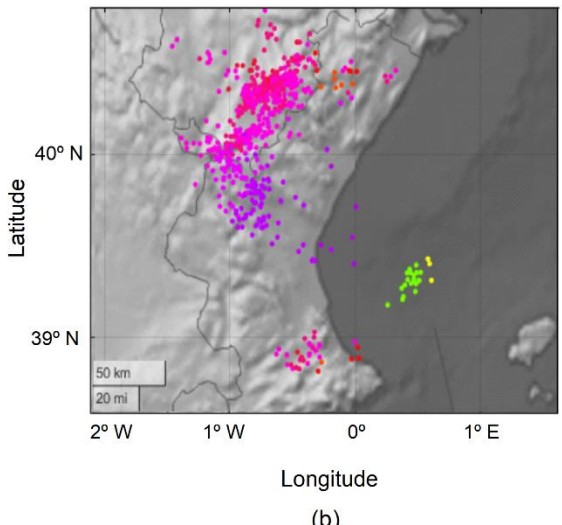

(a)                                                  (b)

**Figure 13: The lightning storm of 17 August 2022 on the Spanish Mediterranean coast. Location of the lightning strokes with circular dots, different colors for different time periods: (a) AEMET. Image source: AEMET_C. Valenciana@AEMET_CValencia. 2022. Accessed via https://x.com/AEMET_CValencia/status/1560254824968785920; (b) WWLLN.**

In our opinion, an acceptable qualitative match is observed, although it must be noted that a rigorous statement on the quality of the results would require a quantitative comparison more than mere image comparisons such as those shown in Figs. 11 to 13. Bearing this subjective and approximate sense in mind, a good reasonably concordance can be appreciated for the point distribution giving location and the color for the corresponding time. The WWLLN network detects fewer lightning strokes than AEMET, because it does not detect low power discharges, showing an important DE decrease below 50 kA, as described by Fig. 8. This is especially noticeable in the Northwest and Northeast areas in Figure 13 where lightning strokes detected by AEMET are not observed in the WWLLN results. On the other hand, WWLLN detects IC lightning that AEMET has discarded, which justifies that discharges detected by WWLLN do not appear in the AEMET map. The qualitative concordance shown in this study indicates that the WWLLN data are a useful tool for thunderstorm tracking which can be used in combination with other techniques (Du et al., 2022). More specifically, Figs. 11 to 13 shows the capability of WWLLN to provide good agreement with LF data from AEMET to resolve convective-storm cells within a larger storm complex generated in a Cut-off Low Pressure System feed with the humidity of the Mediterranean Sea, which is typically a more frequent phenomenon in the Western Mediterranean Basin than in inner continental areas, such as it happens in the Spanish Plateau considered before in this work.

**Conclusions**

The work presented here contributes to the set of existing studies that analyze the operation of the WWLLN around the World, which did not include characteristics of this network in European countries yet. The performance of the WWLLN is evaluated in the area of Spain by comparison with data from the Spanish AEMET network as ground truth during the time period from 1 January 2012 to 30 April 2012, soon after the deployment of a new WWLLN station in Spain. At that moment, sensors in the Spanish area were very close in terms VLF receivers, at a short distance of around 800 km, while typical receivers were between 5000 km and 15000 km in other regions. The current number and distribution of the WWLLN stations, around 70

stations with around 60 active ones, is similar to that considered in the study with data from 2012, therefore, results presented here are valid nowadays although, based on WWLLN growth, it is reasonable to assume that the 2012 DE is a lower bound to the present DE in Spain. Moreover, if the evolution of the AEMET network has surpassed the evolution of the WWLLN, the DE relative to AEMET might even be lower now than in 2012.

For the time interval considered, a global study for the whole region of Spain has been firstly addressed. A total of 20,651 lightning strokes were detected by AEMET in this case. As regards the coincident detections by the WWLLN, 5,904 out of the 20,651 lightning strokes detected by AEMET are also detected by WWLLN in the same area. This yields a theoretical CG detection of 29 % of the lightning strokes detected by the AEMET. The rest of lightning strokes detected by the WWLLN seems to mainly correspond to CC and IC strokes, which are not considered by AEMET. This DE value of 29 % is a

significantly good result for the WWLLN as compared to its behavior for other areas summarized in Table 1. The study of the influence of the lightning peak current on the efficiency and location errors shows results consistent with previous reported works. It is worth noting that the DE considerably improves with high energy strokes, the more relevant to be monitored, with DE values above 50 % when peak currents are higher than 50 kA, approximately.

The previous study for the whole area of Spain has been followed by a subsequent one concerning four reduced regions with different geographical characteristics: a continental homogeneous area, the Spanish Plateau, and three regions including sea and sea-land transitions: the East and South Spanish Mediterranean coasts and the West African Atlantic coast. This second study shows that the qualitative differences in the storms occurring at these different areas can be translated into objective quantitative differences regarding the energy distribution of the lightning strokes, which may explain the differences in the

WWLLN performances. Similar results have been obtained for the accuracy of the four subregions. As regards the efficiency, higher values have been obtained for the results at the coastal regions, especially at those with higher maritime currents. The distribution of peak current of the lightning strokes at these areas shows that the regions with higher values of the DE present an energy distribution with more content at the high energy zone. Since, as shown in Fig. 8, the DE increases with higher energy strokes, it seems reasonable to think that the difference in energy distribution explains the higher values obtained for

the West African Atlantic coast and the South Mediterranean Spanish coast. As regards the relatively high DE value of 29 %

for the whole area of Spain when compared to other more homogeneous regions in Table 1, it seems that the peninsular geographical characteristics of Spain, with important presence of coastal regions compared to inland regions, may be the reason for such high DE value. Concerning, the high variability of the DE for the different studied subregions of Spain, it seems that the rapidly changing geographical characteristics of Spain produced by its peninsular shape are in the basis of this variability, which draws attention on the differences in the expected DE values if important geographical changes are present in other areas.

A final application of the WWLLN shows the global network capabilities to monitor the time evolution of climatic events. The study of three severe storms which affected the Mediterranean Spanish coast at Valencia during years 2020, 2021, and 2022 seems to show a qualitative good agreement with screenshot results available from the AEMET national agency used as reference in this work.

## Competing interests

The contact author has declared that none of the authors has any competing interests.

## Author contribution

E. A. Navarro and J. Segura-García installed the VLF station and are responsible of its maintenance. They also developed the MATLAB codes used for massive data processing.

E. A. Navarro, J.A. Portí, and A. Salinas conceived the analysis, coordinated the work, revised the data and results, and wrote the manuscript.

Sergio Toledo-Redondo participated in analysis, essential manuscript reviews and editing and also provided project resources.

I. Albert, A. Castilla, and V. Montagud-Camps were responsible of collecting data and applying the MATLAB codes to obtain the presented results. They also participated in essential manuscript reviews.

## Acknowledgments

We acknowledge support of MCIN/AEI 10.13039/501100011033 (grant PID2020-112805GA-I00). The authors wish to thank the World Wide Lightning Location Network (http://wwlln.net), a collaboration among over 70 universities and institutions, for providing the lightning location data used in this paper.

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
