# Peer review of "The World Wide Lightning Location Network (WWLLN) over Spain"

_EGUsphere, 2024_

## Referee Comment (RC2)

Review of manuscript 10.5194/egusphere-2024-704

**The World Wide Lightning Location Network (WWLLN) over Spain**

**Summary** This manuscript compares a VLF-based global lightning detection network (WWLLN) to an LF-based local lightning detection network operated by AEMET. The comparisons are made over several regions at the Iberian peninsula, during a 4-month interval in 2012, and also during intense storms on days in 2020, 2021, and 2022. In particular, the stroke detection efficiency (DE) and stroke location accuracy (LA) of WWLLN relative to AEMET strokes were derived. Secondary goals were to examine DE and LA variation with geography, and to compare WWLLN and AEMET strokes during three large storms. A prominent finding was the unexpectedly high (38%) WWLLN detection efficiency for Jan–Apr 2012, over the largest studied region surrounding Spain. This seems to be the first such comparison done for Europe.

**Overall Assessment** At 12 figures and 2 tables, the content of this manuscript is mostly observations, with limited analysis. Observations are reduced to standard statistical quantities and then briefly commented upon. As a contribution to the science literature, the importance of this work is limited by the use of older datasets, and discussion of implications of the results being speculative rather than critical and complete. Nevertheless, as a new comparison of two lightning detection networks, this work is publishable.

**General Comments** The finding of an unusual unexplained high DE of 38% around Spain may indicate an error. Adding to the puzzle, the DE in sub-regions, Fig 2 green and cyan boxes, was smaller. That requires an even higher DE outside the green and cyan boxes to give an average DE over the full region of 38%. Unfortunately, it is not convenient to check the analysis through independent calculations. Perhaps AEMET data had been over-filtered to eliminate weaker CG strokes. Figure 7 is helpful to address this, although there is not enough information about filtering to eliminate this possibility. There may be a problem with how DE is calculated (see comments about section 4).

A number of minor errors should be corrected. Several questions popped up, some due to incomplete descriptions, some some of which may suggest modifications that would improve the paper, and other questions may be outside the scope of this work. When this review was written, no other comments about this manuscript had been viewed; these comments were independently produced.

**Abstract** First sentence is good, but the following text has too much detail for an abstract. Consider deleting sentences after the first down into line 22. Then resume with: *This study finds the detection efficiency of WWLLN is around 38%* ... and continue with the remaining text in the abstract.

**Tables**

Caption on Table 1 declares a date range 2004–2022. However, datasets in the table are from 2003–2015, while publication references are from 2004–2018.

Table 1 has some historical interest, but could be shorter. Much of the contents is not relevant to WWLLN in 2012 or in the 2020s, because algorithms and network station distribution have changed greatly.

**Figures**

*Figure 2*: Text near line 565 describes the cyan region of Figure 2 as the boundary for study, but Figures 10–12 show strokes outside the Figure 2 region. Is the text wrong? Or do Figures 10–12 show strokes not considered in the analysis?

*Figure 5A*: horizontal axis label is wrong. West should be negative, but the label has west is positive.

*Figure 8*: shows a point above DE=1. Isn't that impossible? The method for calculating uncertainties cannot be correct, for an uncertainty bar extending above DE=1 is wrong. The smoothed blue line is difficult to see behind the red circles. If the blue line were plotted on top of the circles, both would be visible.

**Section 1, Introduction**

*lines 69–75*: The method for calculating DE seems to be incorrect (see comments for Section 4).

A CG stroke is not a vertical column above a point on Earth's surface. The path of a stroke often has a large horizontal displacement. Given this behavior of strokes, what is meant by stroke location? For a VLF

stroke detection, this is an effective point for the transmitting antenna location. That point is unlikely to be the stroke contact point on the ground, and it is unlikely to be the effective location of an LF transmitting antenna. Stroke location is expected to be slightly different for different kinds of instruments and it is not defined or meaningful below some distance scale. These considerations mean that one must be careful in finding meaning in stroke accuracy, at small value, from different instruments.

**Section 2, WWLLN**

*lines 91–93, 119–120*: Fig 2 of 10.1029/2020GL091366 (Lightning in the Arctic) shows the history of the count of active WWLLN stations. The number of *active* stations is the important number for network performance, and is always less than the number of stations. Some stations are offline at any time due to network, power, or other technical issues.

*lines 120–126*: For a recent comparison over time, WWLLN detection efficiency compared to New Zealand lightning network is shown in Fig 1 of 10.1029/2019JD030975 (Global Distribution of Superbolts).

Here and elsewhere (lines 416–418, 671–673), is the idea that a higher density of nearby stations might cause DE to be higher. Even assuming station density is high and that DE is high, there is no analysis in the paper showing a cause-effect relation. The link between WWLLN station density around Spain and the apparently higher DE is purely speculation in this paper, and that should be made clear wherever this possible link is mentioned.

Many stations nearby is offered as a possible explanation for the high DE around Spain. However, there are other considerations for stations near a storm that work against this explanation: (1) close strokes can saturate the receiver, distorting the waveforms; (2) close strokes have less frequency dispersion, making it harder to extract the time of group arrival; (3) when several nearby stations see a distant stroke, much of the information is redundant; (4) nearby strokes have more high frequency content that is noise in the VLF analysis—these high frequencies decay quickly with distance. In the stroke location algorithm, stations closer than 300 km to a stroke are not used.

*lines 136–145*: Some details in the list are wrong. No power or peak current estimates are produced in the WWLLN stroke analysis. Instead, stroke VLF (5–18 kHz) energy and uncertainties, in Joules, are output. Stations used for energy calculation are restricted to be in the range 1000–8000 km from the located stroke.

*lines 157–158*: Only stroke VLF energy is provided by WWLLN. An approximate linear relation between stroke VLF energy and peak current is sometimes used to estimate peak current. Of course, once a WWLLN stroke is matched to an AEMET stroke, peak current is available from the AEMET data.

*lines 167–168*: *(DE and LA) must be considered as global value, since they correspond to the detected lightning strokes, independently of their current peak amplitude.* If I understand what this sentence should mean, global is a confusing word to use here. Possible rewording: *(DE and LA) values were calculated from all the matched lightning strokes, independently of their peak current.*

*line 185*: DE in the early days of WWLLN was about 1%, but DE at these early times in the development of the network are not relevant for this study (2012 and 2020–2022). For 2012, WWLLN global DE was probably above 10%. For 2012, the median number of active stations on any day was 55. The total number of stations, 69, is a less useful number.

**Section 3, AEMET**

*line 294*: Are AEMET stroke times rounded to the nearest second or truncated to the second? For instance, if a stroke was originally time-stamped at 01:23:45.678901 at microsecond accuracy, is that stroke's AEMET time published as 01:23:45 or 01:23:46? The method used affects DE.

*lines 299–301*: The description for how AEMET IC and CG strokes were separated is incomplete. How exactly was this done? How sensitive are DE calculations to the criteria used to separate strokes types?

**Section 4, WWLLN performance in Spain**

*lines 406–411*: The calculation of WWLLN DE relative to AEMET appears to be incorrect, in a way that over-estimates DE. The correct DE calculation is to count the number of AEMET strokes that match a WWLLN stroke, then divide by the total number of AEMET strokes. This ratio is the fraction of AEMET strokes also detected by WWLLN—a detection efficiency. The method in the manuscript finds the number of WWLLN strokes that match an AEMET stroke, then divides that by the total number of AEMET strokes. It is a ratio of WWLLN strokes to AEMET strokes, and is not a detection efficiency. It over-estimates DE because it is possible for several WWLLN strokes to match with one AEMET stroke, especially within a 1 second matching time window. Because WWLLN cannot detect the same AEMET stroke more than once, the numerator of the ratio becomes larger than it should be, and DE is over-estimated.

*lines 420–423*: Although there is a resemblance, this is not a Rayleigh distribution. The tail is too heavy; Rayleigh falls fast, as $\exp(-x^2)$. This distribution is also too narrow around its median value. Graphing a few Rayleigh distributions and comparing with the histogram shows this easily. Statistical methods, such as $\chi^2$ goodness-of-fit, would confirm this numerically.

*lines 549–553*: The claim of DE/LA differences in the two geographic regions seems plausible, but there is no evidence or a clear line of reasoning that explains how geography relates to DE/LA differences. For example, a study of the WWLLN stroke energies or the AEMET peak currents in the two regions could explore whether different stroke energy distributions explain the DE differences. The assertion that there are more intense atmospheric phenomena occurring in this region should have a reference. There is only speculation here, with the goal stated in the abstract, lines 25–26, not being met.

**Section 4 Conclusions**

*lines 662–663*: Number of stations in 2012 compared with now: what matters is number of active stations, not number of stations. Both have grown over the years. A correct statement is that based on WWLLN growth, it is reasonable to assume that the 2012 DE is a lower bound to the present DE in Spain. However, if the AEMET network has also improved enough, then WWLLN DE relative to AEMET might even be lower now than in 2012.

*line 688*: How does one decide that there is good agreement between AEMET and WWLLN? This seems like a subjective evaluation, which is ok. But it should be clear this is an opinion rather than a rigorous finding.

**typos/grammar**

- Line 23: considerable→considerably
- Column heading in Table 2: change Est-West→East-West
- Line 284: his→its
- Line 295: Jan 1, 2120–Apr 20, 2012 should be Jan 1, 2012–Apr 20, 2012
- Line 319 change son→soon

---

## Author Comment (AC2)

**Response to comments by reviewer
2 on the manuscript
10.5194/egusphere-2024-704**

**The World Wide Lightning Location Network
(WWLLN) over Spain**

First of all, the authors would like to than the reviewer for his valuable comments, which will help us in improving the quality of the manuscript. We include below a detailed response to the Reviewer's comments. We hope you find this response satisfactory.

We would like to note that, according to the Editor's instructions, the revised manuscript must not be prepared at this stage, so specific changes in the paper and final figures are still pending of the Editor's decision about the further handling of the manuscript.

**Reviewer's General Comments.** *The finding of an unusual unexplained high DE of 38% around Spain may indicate an error. Adding to the puzzle, the DE in sub-regions, Fig 2 green and cyan boxes, was smaller. That requires an even higher DE outside the green and cyan boxes to give an average DE over the full region of 38%. Unfortunately, it is not convenient to check the analysis through independent calculations. Perhaps AEMET data had been over-filtered to eliminate weaker CG strokes. Figure 7 is helpful to address this, although there is not enough information about filtering to eliminate this possibility. There may be a problem with how DE is calculated (see comments about section 4).*

*A number of minor errors should be corrected. Several questions popped up, some due to incomplete descriptions, some of which may suggest modifications that would improve the paper, and other questions may be outside the scope of this work. When this review was written, no other comments about this manuscript had been viewed; these comments were independently produced.*

**Response.** As regards the first paragraph, the reviewer is right in noting the high value of the DE obtained and the strange average value which suggest that other subregions must have even higher values for DE. He is also right in noting the existence of a problem with the way we have calculated DE. As he points in comments to Section 4, there was an error on the original manuscript, since we identified which strokes detected by WWLLN were also detected by the reference agency, AEMET. The correct way was to find which AEMET strokes were also detected by WWLLN. We thank the reviewer for noting this error which may lead to an overestimate in the DE.

The DE calculation has been corrected on the revised manuscript, as well as the corresponding sentences throughout the paper and affected figures which have been modified accordingly. The correction causes an average value reduction from 38% to 29% for the whole region of Spain. As regards the two reduced regions presented in section 4.2 of the original manuscript, the DE reduces from 14.5% to 13% for the Spanish Plateau and from 25% to 22% for the Mediterranean Spanish coast at Valencia.

The reviewer also notes that the surprising high DE global value for Spain indicates that there are regions where the DE must be even higher. In this sense, the revised manuscript will include two new regions with usually high intensity storms. The subregions will be indicated by slightly modifying figure 2 as shown below. The first region (with magenta color in the figure below) corresponds to a region including Canary Islands and the West African Atlantic coast covered by AEMET between [27°N, 37°N]x[20W°, 5°W], while the second one corresponds to the Alborán Sea, [35°N, 37°N]x[5W°, 0°W], at the South Mediterranean coast of Spain (with dark blue color in the figure below). The first region includes a transition between the Atlantic Ocean, while the second one is a transition between land areas and a small sea, the Mediterranean Sea, including the Straits of Gibraltar, a region with frequent strong marine currents. The DE obtained for these regions is 49% and 53%, respectively, which justify the high average new value for Spain of 29%.

[Figure]

Figure 2. Different areas for the studies presented in this work.

According to figure 8 in the original paper showing the DE for different peak amplitudes, the DE value considerably increases with high energy strokes. In this sense, as kindly suggested by the reviewer and in order to try to explain the origin of the differences in the DE for the different subregions considered, the peak distribution of lightning strokes for each subregion has been calculated for this revised manuscript (similar to figure 7 but limited to the four subregions). The resulting figures are shown below.

[Figure]

Figure caption: (new figure 10) Distribution of AEMET return strokes also detected by the WWLLN, in blue color, and total AEMET return strokes, in orange color for different subregions: a) Spanish Plateau, b) East Spanish Mediterranean coast, c) West African Atlantic coast, and d) South Spanish Mediterranean coast.

It can be appreciated from them that the continental area at the Spanish Plateau presents an important distribution of lightning strokes at low energies, while the presence of high energy strokes increases in the other three areas containing land-sea transitions in the following order: East Spanish Mediterranean coast, West Atlantic region, and South Spanish Mediterranean coasts. This, combined with the results shown in Figure 8 in the original manuscript, seems to indicate that the DE is higher in land sea transitions influenced by a different energy distribution towards higher peak currents in the storms for those areas.

The revised manuscript will include the study for the two new subregions in section 4.2, the peak current distribution figure shown above will be included and a discussion on the possible link between this energy distribution and the DE values will be addressed.

**Reviewer's Specific Comments**

*Abstract  First sentence is good, but the following text has too much detail for an abstract. Consider deleting sentences after the first down into line 22. Then resume with:  This study finds the detection efficiency of WWLLN  is around 38% . . . and continue with the remaining text in the abstract.*

**Response**

The paragraph will be reduced and reorganized to eliminate the excessive details but still providing a brief introduction of the WWLLN to researchers non-directly concerned with this global network.

*Tables*

*Caption on Table 1 declares a date range 2004–2022. However, datasets in the table are from 2003–2015, while publication references are from 2004–2018.*

*Table 1 has some historical interest, but could be shorter.  Much of the contents is not relevant to WWLLN in 2012 or in the 2020s, because algorithms and network station distribution have changed greatly.*

**Response**

The caption has been corrected. As regards the reduction of contents, we think that the items included not only describes the historical evolution of the network performances, showing the differences in the working parameters and resulting DE for different studies, but also facilitates the understanding of the paragraphs describing the WWLLN feature evolution since its initial times (lines 160 and following in the original manuscript).

In this sense and pointing to the interest in including summarizing details in the table, Reviewer 1, in one of his comments, makes reference to a detail in the text that is not in the Table ( *L190: "...the best data recorded by WWLLN so far was a DE of 31%...". This value is not even in your Table 1*). It seems that summarizing capability of this table advises maintaining information even when it could be considered as non-relevant for highly specialized readers.

**Figures**

**Figure 2:** *Text near line 565 describes the cyan region of Figure 2 as the boundary for study, but Figures 10–12 show strokes outside the Figure 2 region. Is the text wrong? Or do Figures 10–12 show strokes not considered in the analysis?*

**Response.** The reviewer is right. The region only approximately corresponds to the cyan region of the study un section 4.2. It has been chosen to match the areas covered by the maps provided by AEMET. The text will be changed accordingly and the areas of the original figures 10b, 11b, and 11c will be adjusted to match the regions of AEMET shown in figures 10a, 11a, and 12a, respectively.

**Figure 5A:** *horizontal axis label is wrong. West should be negative, but the label has west is positive.*

**Response.** Thank you for noting the mistake. The figure caption will be corrected.

**Figure 8:** *shows a point above DE=1. Isn't that impossible? The method for calculating uncertainties cannot be correct, for an uncertainty bar extending above DE=1 is wrong. The smoothed blue line is difficult to see behind the red circles. If the blue line were plotted on top of the circles, both would be visible.*

**Response:**

As regards the point above DE=1, it was due to the mistake in calculating the DE the reviewer mentions in section 4, which provides an overestimate of the DE. This calculation has been corrected and the DE values are now below 1 as expected.

As regards uncertainties, the values above DE=1 result from direct statistical operations using the set of available data, which may lead to unphysical solutions. Of course, these statistical operations must be completed with the condition that DE is lower or equal than unity. In the figure of the revised manuscript, the vertical axis has been redefined to avoid unphysical solutions and the blue line has been plotted on top of the circles to better appreciate it.

**Reviewer's Comments on Section 1, Introduction**

**Comment:** *lines 69–75: The method for calculating DE seems to be incorrect (see comments for Section 4).*

*A CG stroke is not a vertical column above a point on Earth's surface. The path of a stroke often has a large horizontal displacement. Given this behavior of strokes, what is meant by stroke location? For a VLF stroke detection, this is an effective point for the transmitting antenna location. That point is unlikely to be the stroke contact point on the ground, and it is unlikely to be the effective location of an LF transmitting antenna. Stroke location is*

*expected to be slightly different for different kinds of instruments and it is not defined or meaningful below some distance scale. These considerations mean that one must be careful in finding meaning in stroke accuracy, at small value, from different instruments.*

**Response.** As we mention above, the DE calculation has been corrected in the sense indicated by the reviewer's comments on section 4 and all related text, figures, tables…, have been modified accordingly.

As regards the care that must be taken when talking about location accuracy of a lightning stroke, a sentence clarifying the difficulty in defining the lightning location will be included in section II after line 117 in the original manuscript, where lightning location is described. The paragraph will read as follows:

> "… is simultaneously detected by a minimum of 5 stations. In addition to the above-mentioned difficulties, it is worth noting that care must be taken when interpreting data of lightning location below some length scale. This is so because the distance determined by WWLLN corresponds to an equivalent VLF antenna transmitting from an effective point, but the actual lightning stroke path is not usually a vertical path, thus the distance detected will not exactly coincide with the stroke contact point on the ground. The meaning of this approximate distance is even more approximate because of the comparison with independent AEMET results, obtained with LF technology, where characteristic distances differ from those of WWLLN."

**Reviewer's comments on Section 2, WWLLN**

*Comment: Lines 91–93, 119–120: Fig 2 of 10.1029/2020GL091366 (Lightning in the Arctic) shows the history of the count of active WWLLN stations. The number of active stations is the important number for network performance, and is always less than the number of stations. Some stations are offline at any time due to network, power, or other technical issues.*

**Response:** The sentence at line 91-93 in the original manuscript will be modified mentioning the active stations and a reference to *10.1029/2020GL091366* will be added in the following terms:

> "The distribution of associated active stations around the whole Earth, slightly above 60 stations since 2014 (Holzworth et al., 2021), makes the WWLLN achieve global location of lightning strokes at a planetary scale with a constantly improved accuracy…"

The sentence in line 119-120 will be modified in similar terms:

> "The number of active stations increased until an almost stable number around 60 active stations since 2014, approximately."

*Comment: Lines 120–126: For a recent comparison over time, WWLLN detection efficiency compared to New Zealand lightning network is shown in Fig 1 of 10.1029/2019JD030975 (Global Distribution of Superbolts).*

**Response:** The paragraph beginning in line 119 will include a sentence with the recent comparison for New Zealand and the corresponding reference. The paragraph will read as follows:

> "The WWLLN had 40 receiving sensors in 2010, providing a DE~11% in 2010 for peak currents greater than 20 kA (Abreu et al., 2010) and a LA of around 5 km. A recent comparison over time of the WWLLN detection efficiency for different peak currents can be found in (Holzworth et al., 2019) for the New Zealand area…."

*Comment: Here and elsewhere (lines 416–418, 671–673), is the idea that a higher density of nearby stations might cause DE to be higher. Even assuming station density is high and that DE is high, there is no analysis in the paper showing a cause-effect relation. The link between WWLLN station density around Spain and the apparently higher DE is purely speculation in this paper, and that should be made clear wherever this possible link is mentioned.*

*Many stations nearby is offered as a possible explanation for the high DE around Spain. However, there are other considerations for stations near a storm that work against this explanation: (1) close strokes can saturate the receiver, distorting the waveforms; (2) close strokes have less frequency dispersion, making it harder to extract the time of group arrival; (3) when several nearby stations see a distant stroke, much of the information is redundant; (4) nearby strokes have more high frequency content that is noise in the VLF analysis—these high frequencies decay quickly with distance. In the stroke location algorithm, stations closer than 300 km to a stroke are not used.*

**Response:** The reviewer is right. The nearest stations are probably saturated when high energetic events happen (maybe this is not the case for the low intensity case, but the reviewer is right noting that the sentence was speculative). In addition, since we have no way of proving the link between the higher density of stations in Spain and the high results for DE, we will eliminate those unproved ideas from the paper.

*Comment: lines 136–145: Some details in the list are wrong. No power or peak current estimates are produced in the WWLLN stroke analysis. Instead, stroke VLF (5–18 kHz) energy and uncertainties, in Joules, are output. Stations used for energy calculation are restricted to be in the range 1000–8000 km from the located stroke.*

**Response:**

Dear reviewer you are right. The readme.txt file gives the format of files:

%The format for the APP files is:

%YYYY/MM/DD, hh:mm:ss, lat, long, uncertainty (µs), nstn, power (kW), power,…
uncertainty (kW), nstn_power.

No power or peak currents are estimated, instead power in kW and power uncertainty in kW are given. This will be corrected in lines 136-145.

**Comment:** *Lines 157–158: Only stroke VLF energy is provided by WWLLN. An approximate linear relation between stroke VLF energy and peak current is sometimes used to estimate peak current. Of course, once a WWLLN stroke is matched to an AEMET stroke, peak current is available from the AEMET data.*

**Response:** Thank you for noting the error. The revised manuscript will eliminate the sentence from lines 157-158 and a sentence will be added at line 474 from the original manuscript to clarify that the peak current of the WWLLN strokes are those corresponding to the matched AEMET strokes. The paragraph will read as follows:

> "… strokes in the AEMET data, a 16.8% of the detected CG strokes. As regards the WWLLN, the peak current is not provided by this network, thus, each detected stroke is assigned the peak current corresponding to the matched AEMET stroke. The same ratio is preserved for …."

**Comment:** *lines 167–168: (DE and LA) must be considered as global value, since they correspond to the detected lightning strokes, independently of their current peak amplitude. If I understand what this sentence should mean, global is a confusing word to use here. Possible rewording: (DE and LA) values were calculated from all the matched lightning strokes, independently of their peak current.*

**Response:** This comment is also made by the first reviewer. We agree that WWLLN is a global network and the term global for DE may not be appropriate. We have changed "global" by "total detection values", and slightly modified the sentence in the following terms:

> "It is worth noting that these two quantities must be considered as total detection values, i.e., they correspond to the detected lightning strokes, independently of their current peak amplitude. More detailed information on the DE values for specific current peak amplitudes can be found in the works referenced in Table 1"

**Comment:** *line 185: DE in the early days of WWLLN was about 1%, but DE at these early times in the development of the network are not relevant for this study (2012 and 2020–*

*2022). For 2012, WWLLN global DE was probably above 10%. For 2012, the median number of active stations on any day was 55. The total number of stations, 69, is a less useful number.*

**Response:** We think that the historical review of WWLLN state of development and performances may be useful for interested readers, but the reviewer is right in noting that this low value corresponds to the early stage of WWLLN and that it soon improved the value above 10% around 2012. The sentences will be modified throughout the text to clarify his point.

**Reviewer's comments on Section 3, AEMET**

*Comment: line 294: Are AEMET stroke times rounded to the nearest second or truncated to the second? For instance, if a stroke was originally time-stamped at 01:23:45.678901 at microsecond accuracy, is that stroke's AEMET time published as 01:23:45 or 01:23:46? The method used affects DE.*

**Response:** We have no information on that aspect.

*Comment: lines 299–301: The description for how AEMET IC and CG strokes were separated is incomplete. How exactly was this done? How sensitive are DE calculations to the criteria used to separate strokes types?*

**Response:** Dear reviewer in lines 301-311 we give references where this issue is addressed. The reader can go further reading these references. We assume that AEMET data is CG. This was explained also to reviewer 1. The WWLLN data include both IC and CG. From our knowledge there is a ratio IC/CG that is obtained doing special surveys using videos of the storms.

**Reviewer's comments on Section 4, WWLLN performance in Spain**

*Comment: lines 406–411: The calculation of WWLLN DE relative to AEMET appears to be incorrect, in a way that over-estimates DE. The correct DE calculation is to count the number of AEMET strokes that match a WWLLN stroke, then divide by the total number of AEMET strokes. This ratio is the fraction of AEMET strokes also detected by WWLLN—a detection efficiency. The method in the manuscript finds the number of WWLLN strokes that match an AEMET stroke, then divides that by the total number of AEMET strokes. It is a ratio of WWLLN strokes to AEMET strokes, and is not a detection efficiency. It over-estimates DE because it is possible for several WWLLN strokes to match with one AEMET stroke, especially within a 1 second matching time window. Because WWLLN cannot*

*detect the same AEMET stroke more than once, the numerator of the ratio becomes larger than it should be, and DE is over-estimated.*

**Response:** Thank you for noting this error. As we mention in our previous response to your general comments, the DE has been recalculated in the correct way you indicate: finding which AEMET strokes were also detected by WWLLN.

The following table shows the result obtained for Spain and four the subregions: the Spanish Plateau, the West Mediterranean coast, and the two new subregions considered, namely, the South Mediterranean coasts and West Atlantic region. The table shows the values obtained with the original method (incorrect) and the corrected one. The revised manuscript will include the correct values of the five regions in a new corrected Table 2 and throughout the text.

Table 2 for the revised manuscript

| | DE (%) in the original manuscript | DE after correction (%) | Latitud/Longitude limits |
|---|---|---|---|
| Spain | 38 | 29 | [27ºN, 44ºN]/[18ºW,4.7ºE] |
| Spanish Plateau | 14.5 | 13 | [38ºN, 42ºN]/[6ºW,1ºW] |
| East Mediterranean coast | 25 | 22 | [37.5ºN, 41ºN]/[1ºW,1ºE] |
| West Atlantic region | 67 | 49 | [27ºN, 37ºN]/[18ºW,5ºW] |
| South Mediterranean coast (Alborán Sea) | 79 | 53 | [35ºN, 37ºN]/[5ºW,0ºW] |

*Comment: lines 420–423: Although there is a resemblance, this is not a Rayleigh distribution. The tail is too heavy; Rayleigh falls fast, as exp(−x2 ). This distribution is also too narrow around its median value. Graphing a few Rayleigh distributions and comparing with the histogram shows this easily. Statistical methods, such as χ2 goodness-of-fit, would confirm this numerically.*

**Response.** The resemblance is only qualitative and does not introduce valuable information, therefore the following sentence will be removed:

"It resembles a clear Rayleigh distribution, a distribution typical for nonnegative-valued random variables. This distribution is often observed when the over-all

magnitude values are related to two independent components. This is our case, where the location error depends on two parameters, latitude and longitude"

*Comment: lines 549–553: The claim of DE/LA differences in the two geographic regions seems plausible, but there is no evidence or a clear line of reasoning that explains how geography relates to DE/LA differences. For example, a study of the WWLLN stroke energies or the AEMET peak currents in the two regions could explore whether different stroke energy distributions explain the DE differences. The assertion that there are more intense atmospheric phenomena occurring in this region should have a reference. There is only speculation here, with the goal stated in the abstract, lines 25–26, not being met.*

**Response:** The reviewer is right in noting the speculative nature of our statement. It is based on our particular knowledge of differences in storms in Spain, the origin of the authors, but this is not a proper reference to support such a statement. The reviewer wisely proposes a study of stroke energy distribution for each region. As we mention when answering the general reviewer's comments, we have done this study. A new figure 10 (see page 3 of this response) will be included and discussed. It can be seen from this figure a difference in the energy distribution: regions with more energetic strokes correspond to regions with higher DE values. The new figure and the possible explanation for the differences in DE values will be included in the revised manuscript.

**Section 4 Conclusions**

*Comment: lines 662–663: Number of stations in 2012 compared with now: what matters is number of active stations, not number of stations. Both have grown over the years. A correct statement is that based on WWLLN growth, it is reasonable to assume that the 2012 DE is a lower bound to the present DE in Spain. However, if the AEMET network has also improved enough, then WWLLN DE relative to AEMET might even be lower now than in 2012.*

**Response:** Thank you for the comment. We will include a sentence after line 663 about this comment.

*Comment: line 688: How does one decide that there is good agreement between AEMET and WWLLN? This seems like a subjective evaluation, which is ok. But it should be clear this is an opinion rather than a rigorous finding*

*Response:* The reviewer is right, we will change the sentences to clarify that this is a qualitative comparison based on visual comparison, not an objective quantitative one.

The typos/grammar will be corrected. Thank you for letting us know.

---

## Author Response (AR1)

**Response to comments by Reviewer 1 to the manuscript:**

egusphere-2024-704
Title: The World Wide Lightning Location Network (WWLLN) over Spain
Author(s): Enrique A. Navarro et al.
MS type: Research article

First of all, the authors would like to thank the reviewer for his/her valuable comments, which we hope will help us in improving the quality of the manuscript.

The reviewer comments are split into general concerns and specific remarks. In the following paragraphs, you can find the response to all the concerns and remarks.

Thank you again for your valuable comments.

**RESPONSE TO GENERAL CONCERNS**

**Concern 1:**

*- The data spans only a four-month period (January 1 – April 30, 2012) and is outside the typical thunderstorm season, encompassing a total of only 20,651 lightning strokes within the yellow polygon (and drops further to 3389 strokes in the green square, and an even lower amount of detections within the cyan rectangle which is not being mentioned). The rationale provided is that the AEMET dataset for that time was available as open data.*

*Given that it is now 2024, a more recent dataset would be significantly more valuable. This is not only because it would be more current but also because the AEMET network from 2012 differs from the present-day network. The justification for using this specific dataset, as mentioned in lines 337-339, is insufficient. The authors acknowledge the WWLLN network for providing the lightning location data used in this paper. I don't see why the authors cannot ask a more recent data set to AEMET. Moreover, the time resolution within this open dataset is only one second, making it unusable for a comparison study.*

**Response**:

We agree with the reviewer that using more recent reference data would be more appropriated, but we have tried to contact AEMET asking for historical data in order to carry the study and we have had no response. Especially difficult is the access to the historical data usually required for scientific purposes. This difficulty may suggest that the existence of other lightning networks, such as WWLLN, with more available historical data, seems then of great interest for the scientific community. It is worth noting that the AEMET and the WWLLN stations have different objectives. The AEMET distribution is intended for a more detailed monitoring of meteorological events happening in Spain, with a dense distribution of stations. As regards the WWLLN, it is a global network intended to provide data to support global studies by using a world-wide low-density distribution of VLF stations. In this sense, the aim of

the manuscript is not a complete and deep comparison between AEMET stations and WWLLN stations to determine which of the two networks is the best, but a reasonably good test of the WWLLN performances in order to stablish its capability for detecting lightning activity in the area considered, Spain and surrounding areas, which has not been reported yet in the specialized literature.

Bearing in mind the above-mentioned difficulty in accessing the data from national or regional meteorological services, the relevant question concerning the manuscript under consideration is whether the use of data from 2012 invalidates or reduces the conclusions of the work presented in the manuscript. As mentioned above, there is no study that considers the performances of the WWLLN at western Europe yet, despite this study has been carried out and reported for other areas of the world. Since WWLLN is a global network, we think that the study is pertinent and relevant. The conclusions by comparing data from same time periods, 2012, seems fair and enough to give values of detection efficiency (DE) and location accuracy (LA). The difficulty in measuring lightning activity causes that values already reported in the literature have great error estimations (see Table I in the paper and answer to a comment below in this response). This makes that possible differences that may arise from technical improvements implemented since then in both networks are not so relevant and the conclusions of the presented work, comparing data from WWLLN and AEMET corresponding to 2012, can be considered as still valid in present days and, therefore, we think that the study in this manuscript represents a valid prior calibration stage which can be considered as a lower bound of the present WWLLN performances.

Another important point we would like to note is that in the scientific community, and more specifically in the communities involved in the study of natural phenomena, it is common the study of past phenomena, often occurred several years ago, from a new point of view. In this case, the fact the data may be old might not be as relevant as the data existence and their accessibility. In this sense, the data made public by AEMET are very useful for a study of the present atmospheric phenomena, but we think that WWLLN provides a relatively accessible database of historical past data, which may be useful for scientific studies involving meteorological phenomena during large time spans, as long as these data are duly verified. We think that the work we present contributes to this data calibration study.

In addition to the initial work using 2012 data, a subsequent study of more recent activity has also been addressed and included in the manuscript. The activity considered corresponds to three large storms that generated risks of flooding and hail for the population and mainly putting the Valencian agricultural production at risk: April 2020, August 2021, and August 2022. We have tried to contact AEMET to ask for recent data but we have had no response from the agency. The existence of other global networks with more accessible historical data seems then an important benefit. For the study, the screenshots of the available figures in the AEMET website have been used. The results shown in Figures 11 to 13 of the revised paper show a qualitatively good behavior.

In any case the reviewer is right in noting that a study with more recent data would be more valuable and we have modified some sentences in the modified manuscript. Concretely, the following sentence has been added in line 320 of the revised version:

"…Currently, the availability for the research community of data from this and other national agencies is usually very difficult or expensive, what increases the interest in having more easily available data from other networks, such as the WWLLN studied here.",

the first paragraph in section 4.1 includes the following sentences:

"… First, data from AEMET were openly available to the authors at that period, and second, this period was close in time to the moment at which the station in Valencia (Spain) was deployed by the team of this work in the year 2011. It must be noted that the objective of this paper is not comparing AEMET and WWLLN in order to determine which is the best network, since they have different local and global objectives, respectively. The main goal is to determine the DE of the WWLLN in the whole region of Spain, which includes continental and insular regions, using the AEMET data as true data (Abarca, 2010). This study is the first one analyzing the WWLLN performances at Western Europe, specifically at Spain, where large geographical differences happen at relatively low distances. The use of data from 2012 does not invalidate the conclusions of the first study, although, bearing in mind the technical improvements of both WWLLN and AEMET since 2012, the results presented here must be considered as a lower bound for the current network performances, which, most likely, will have been improved at present days. ",

and the first paragraph at the conclusions section is modified as follows:

"…. The current number and distribution of the WWLLN stations, around 70 stations with around 60 active ones, is similar to that considered in the study with data from 2012, therefore, results presented here are valid nowadays although, based on WWLLN growth, it is reasonable to assume that the 2012 DE is a lower bound to the present DE in Spain. Moreover, if the evolution of the AEMET network has surpassed the evolution of the WWLLN, the DE relative to AEMET might even be lower now than in 2012."

As regards the time resolution of 1 s, a smaller time resolution would allow using a criteria of time coincidence without combining with a spatial one. This is done in Abreu (2010) and Rodger et al. (2006) who use time resolution of 0.5 ms. The use of coarse resolution must combine time and spatial coincidence as done in several works shown in Table 1. In this sense, this resolution of 1 s is similar to that used in (Fan et al., 2018) and (Kigotsi et al.,2018), where the criterion to set a coincidence is 0.5 s and a distance lower than 50 km, while 0.5s and 20 km are used in (Abarca et al., 2010). In addition, it is worth noting that the typical duration of a flash is in the order of 0.5 s, while several strokes usually happen during a flash, lasting only about 10-20 ms each stroke. As reported by Jacobson (2006), referenced in the paper, each WWLLN station has a triggering system with and average intertrigger time of 0.2 s, which causes that WWLLN detection of a lightning stroke only happens once during a flash and stroke detection becomes basically a flash detection. Since the typical duration of a flash is around 0.5 s, the resolution of 1 s is enough to compare the lightning activity of

AEMET, corresponding to flashes, to the lightning strokes detected by WWLLN, in practice also corresponding to flashes for the reasons stated above.

**Concern 2.**

*The manuscript is submitted to NHESS, which is dedicated to research on natural hazards and their consequences. Although the current manuscript deals with lightning observations, it is essentially a comparison study. Therefore, the more appropriate journal for such a type of study would be EGU Atmospheric Measurement Techniques, which includes intercomparison research.*

**Response:**

We thank the reviewer for this suggestion. It is certainly right in noting that the paper could have been sent to EGU Atmospheric Measurement Techniques, but we still think that the paper is also appropriated for NHESS.  As we mention above, the paper is not exactly about a comparison between AEMET and the WWLLN, since they have different objectives and characteristics, but on studying and presenting the WWLLN performances as a global network. Of course, these performances are assessed by means of a comparison, but the comparison to determine which network is the best is not the final goal of the paper. Bearing this in mind, we think that the content of the paper falls within the second item of the NHESS journal scopes published in the Journal web page (https://www.natural-hazards-and-earth-system-sciences.net/about/aims_and_scope.html):

> o *"The detection, monitoring, and modelling of natural phenomena, and the integration of measurements and models for the understanding and forecasting of the behaviour and the spatial and temporal evolution of hazardous natural events as well as their consequences"*.

The fact that the Editor of NHESS has admitted the manuscript as a preprint seems to support this choice.

**RESPONSE TO SPECIFIC REMARKS**

**Specific remark**:

*In the abstract and the text, it should be clarified whether the DE pertains to strokes or flashes.*

**Response:**

Thank you for the remark. The reviewer is right in noting that a certain ambiguity is present in this sense. The WWLLN identifies lightning activity through very low frequency radiation originated by lightning strokes. In this sense, the WWLLN does not detect flashes, but lightning strokes, instead. However, as described in (Jacobson et al., 2006), referenced in the paper, each WWLLN station has a triggering system with and average intertrigger time of 0.2s, which causes that WWLLN detection of a lightning stroke only happens once during a flash and stroke detection becomes effectively a flash detection. This coincidence in the detection of a flash and a particular stroke of that flash may explain that the slight undefinition that the reviewer has found in our paper is also present in other similar papers referenced in the manuscript. In this sense, and only

as examples of the different terminology: Jacobson talks about lightning or lightning evens in (Jacobson et al., 2006), Abarca uses the term flashes (Abarca et al., 2010) and Abreu uses the term lightning strokes (Abreu et al., 2010), while Thomson mentions "LIS groups that have a coincident with a WWLLNN or ENTLN stroke" (Thomson et al., 2014).

In any case, the reviewer is right and the specific reference to lightning strokes has been made in the revised manuscript.

**Specific remark**:

*In several parts of the text, it is stated that the primary objective of regional and national lightning location networks is to detect CG strokes, with IC events being of lesser importance. I disagree with this. For example, air traffic controllers are very interested in IC activity and usually receive the observations through a LLS network provider.*

**Response:**

The reviewer is right in noting that.

The first sentence in the introductory section has been modified in the following sense and reference (Thomas et al., 2001) has been included:

> "An important objective of regional and national lightning location networks is the detection and tracking of Cloud-to-Ground (CG) lightning strokes. The CG lightning strokes coexist with Cloud-to-Cloud (CC) and Intra-Cloud (IC) discharges. While the study of IC events is of great interest because they are considered the more important natural source of high frequency and very high frequency radiation (Thomas et al., 2001) and have a direct interest for air traffic controllers, for instance, the social interest in monitoring and detecting CG activity relies on the fact …"

Lines 200-201 in the original manuscript have been modified in the following sense (now lines 216-217):

> "As regards the national and regional networks used as reference, they are devoted to the detection of both CG strokes and CC/IC strokes"

**Specific remark**:

*The introduction mentions another global network, ENTLN. For completeness, the authors should also include GLD360.*

**Response:**

Thank you for the suggestion. The network has been added in the revised version (line 42).

**Specific remark**:

*L74: 'excellent' DE. A DE of 38% is far from excellent.*

**Response:**

The reviewer is right. The DE and LA are significantly better than those previously reported in other areas, with DE values below 10%, but "excellent" was inappropriately exaggerated. The sentence in line 72 of the revised manuscript has been changed in the revised manuscript in the following sense:

> "Our work will show that the WWLLN provides values for the DE and LA in the area of Spain which are higher than those reported up to the moment, with remarkable results for high peak current lightning strokes..."

**Specific remark**:

*L166: '...quantities must be considered as global values...': what is meant by 'global' here?*

**Response:**

As explained in the original sentence, the term global refers to the fact that the DE corresponds to the detection of a lightning stroke, regardless of its amplitude. But the detection capability improves with the peak amplitude. A reference to the network performance at different specific amplitudes is also made in the sentence. Maybe using the term global is not a good choice, since the global adjective is also used to define WWLLN as a global network. In this sense, the sentence in line 179 of the revised manuscript has been changed in the following terms.

> "It is worth noting that these two quantities must be considered as total detection values, i.e., they correspond to all the detected lightning strokes, independently of their current peak amplitude. More detailed information on the DE values for specific current peak amplitudes can be found in the works referenced in Table 1."

The term "global" has also been changed to "total" in line 228 for the same reason.

**Specific remark**:

*L190: "...the best data recorded by WWLLN so far was a DE of 31%...". This value is not even in your Table 1.*

**Response:**

The value corresponds to a reduced area of the larger western hemisphere region considered in the work by (Rudlosky and Shea., 2013), which is included in Table I. The sentence in line 205 of the revised version has been changed to clarify this point in the following terms:

> "The best data recorded by WWLLN so far was a DE of 31 % obtained in the Pacific Ocean in January 2010, a reduced area of the whole Western Hemisphere region considered in (Rudlosky and Shea., 2013) and shown in Table 1."

**Specific remark**:

*L197: '...some of them reporting a DE with errors assumed to be between 80 %-90 %'. Explain in more detail what you mean.*

**Response:**

There is a mistake in the original manuscript. Line 197 (now 213) should say "a LA with errors…" instead of "a DE with errors…". We thank the reviewer for making us note the mistake.

Bearing this mistake in mind, the difficulty of measuring lightning activity and knowledge of "ground truth" causes that errors may be comparable to the measured values. As a simple example, let us note that the first reported value of LA in the table by Lay et al. in 2004 is of 20.25±13.5 km, i.e., the error is around 67 % of the average value. Similarly, the LA value of Abreu et al. 2010 is of 7.24±6.24, which means that the error is around 86 % of the value.

In addition, as noted by Reviewer 2, the difficulty in determining the exact position of a lightning strokes is greatly affected by two facts: i) the path of the lightning stroke is not a vertical one and the estimated distance corresponds to the distance of the equivalent antenna substituting the stroke and ii) the combination of VLF technology with LF technology from AEMET makes the typical distances to differ. In this sense, the following sentences have been added (lines 124 to 129 of the revised version):

> "It is worth noting that care must be taken when interpreting data of lightning location below some length scale. This is so because the distance determined by WWLLN corresponds to an equivalent VLF antenna transmitting from an effective point, but the actual lightning stroke path is not usually a vertical one, thus, the distance detected will not exactly coincide with the stroke contact point on the ground. The meaning of this distance is even more approximate because of the comparison with independent AEMET results, obtained with Low Frequency (LF) technology, where characteristic distances differ from those of WWLLN."

**Specific remark:**

*L199-201: '...and make very coarse estimations of CC/IC strokes'. I disagree. There has been significant progress over the past decade in detecting IC/CC events using ground-based LF sensors. This is true, for example, for the NLDN and EUCLID networks.*

**Response:**

Thank you for your right comment. The revised version eliminates "and make very coarse estimations of CC/IC strokes" from the sentence in lines 216-217 of the revised version.

**Specific remark:**

*L248: 'the reference station: the AEMET': AEMET is not an LLS; it is a National Meteorological Service (NMS).*

**Response:**

Effectively, AEMET is not an LLS, but a National Meteorological Service (NMS). The title of Section 3 (line 282) has been changed to:

"3 The reference regional lightning detection system of the Spanish National Meteorological Service, AEMET".

**Specific remark**:

*L256: A small part of Vaisala's business is devoted to lightning detection. The sentence at L256/257 implies that Vaisala's only/main focus is on lightning detection.*

**Response:**

We have omitted that part of the sentence to avoid suggesting that Vaisala's activity is only devoted to lightning detection. The final sentence at lines 288-290 mentions the sensors, that they are manufactured by Vaisala and only adds the link to Vaisala website without mentioning any particular activity of Vaisala.

**Specific remark**:

*L279: Meteo-France does not own the sensors in France. Meteorage is the network provider that sells data to Meteo-France.*

**Response:**

We would like to thank the reviewer for the information. The sentence in original line 279, now in line 307, has been changed in the following terms:

"… and from sensors of Meteorage who provides data to the French meteorological service (Météo-France) "

**Specific remark**:

*L283: Why mention Nunez et al. (2019) when you only use data from 2012? Additionally, this article is only in Spanish and not an international publication.*

**Response:**

We have deleted the reference Nunez et al. (2019) in the revised manuscript. Specific sentences requiring that reference have also been removed (see lines 36, 282, and 304 in the original manuscript, lines 34, 310, and 332 in the revised version).

**Specific remark**:

*L294: '...together with information about the current for the first stroke.' Does this mean only the peak current of the flash is provided and not of all strokes within the flash? This affects Figure 8 and all related discussions about DE variation as a function of peak current.*

**Response:**

As we mention above, a flash event includes several lightning strokes. The first stroke is usually the more energetic one and its peak current provided by AEMET. This peak current from AEMET corresponds to that obtained for a lightning stroke detected by WWLLN and the Figure 8 and subsequent discussion is carried out basing on this comparison.

As we state in line 510 of the revised manuscript, the results of Fig. 8 are very similar to previous works referred to in Table 1 (Abarca et al., 2010; Rodger et al., 2006; Fan et al., 2018), which seems to support the bondage of this procedure and subsequent discussion.

**Specific remark**:

*Sect. 4.3: The authors now show AEMET observations of three more recent days. Does this mean the authors have contacted AEMET for more recent data, or are these figures simply screenshots from the AEMET website?*

**Response:**

We have tried to contact AEMET for recent data and we have had no response. The figures from AEMET are screenshots from the AEMET website to make a comparison of recent data. The comparison is then a qualitative one but we think that it is still relevant. We have changed the sentence to clarify that the results presented are based on a comparison with screenshots from AEMET. The sentence is now (line 626):

> "Figures 11 to 13 show the results for the three storm events. Figures 11a, 12a, and 13a are screenshots taken directly from AEMET website, https://www.aemet.es/, while Figs. 11b, 12b, and 13b have been generated with WWLLN data"

**Specific remark**:

*Figure 11 (Fig. 12 in the revised version): The regions displayed in figures a and b should be exactly the same, which is currently not the case.*

**Response:**

The reviewer is right in noting that. In fact, the difference is also present in the original Figures 10 and 12. Figures 11 to 13 in the revised version have been corrected.

**Specific remark**:

*Fig 12 (now Fig. 13): How do you explain the observations in the Northwest and Northeast observed by AEMET that are not present in the WWLLN dataset?*

**Response:**

Thank you for noting that detail which went unnoticed to us. According to the results depicted in Fig. 8, those observations at Northwest and Northeast most likely correspond to low amplitude lightning strokes. Although the sentence at line 583 (now line 696) already mentioned the fact that some lightning strokes were not detected by WWLLN in the following terms:

> "The WWLLN network detects fewer lightning strokes than AEMET, because it does not detect low power discharges, showing an important DE decrease below 50 kA, as described by Fig. 8.",

we have included a sentence at line 698 of the revised version noting this difference specifically for the Northwest and Northeast areas, in the following terms;

"This is especially noticeable in the Northwest and Northeast areas in Figure 13 where lightning strokes detected by AEMET are not observed in the WWLLN results."

**Response to comments by Reviewer 2 to the manuscript:**

egusphere-2024-704
**Title: The World Wide Lightning Location Network (WWLLN) over Spain**
**Author(s): Enrique A. Navarro et al.**
**MS type: Research article**

First of all, the authors would like to thank the reviewer for his/her valuable comments, which we hope will help us in improving the quality of the manuscript. We include below a detailed response to the Reviewer's comments. We hope you find this response satisfactory.

**Reviewer's General Comments.** *The finding of an unusual unexplained high DE of 38% around Spain may indicate an error. Adding to the puzzle, the DE in sub-regions, Fig 2 green and cyan boxes, was smaller. That requires an even higher DE outside the green and cyan boxes to give an average DE over the full region of 38%. Unfortunately, it is not convenient to check the analysis through independent calculations. Perhaps AEMET data had been over-filtered to eliminate weaker CG strokes. Figure 7 is helpful to address this, although there is not enough information about filtering to eliminate this possibility. There may be a problem with how DE is calculated (see comments about section 4).*

*A number of minor errors should be corrected. Several questions popped up, some due to incomplete descriptions, some of which may suggest modifications that would improve the paper, and other questions may be outside the scope of this work. When this review was written, no other comments about this manuscript had been viewed; these comments were independently produced.*

**Response.** As regards the first paragraph, the reviewer is right in noting the high value of the DE obtained and the strange average value which suggest that other subregions must have even higher values for DE. He is also right in noting the existence of a problem with the way we have calculated DE. As he points in comments to Section 4, there was an error on the original manuscript, since we identified which strokes detected by WWLLN were also detected by the reference agency, AEMET. The correct way was to find which AEMET strokes were also detected by WWLLN. We thank the reviewer for noting this error which may lead to an overestimate in the DE.

The DE calculation has been corrected on the revised manuscript, as well as the corresponding sentences throughout the paper and affected figures which have been modified accordingly. The correction causes an average value reduction from 38% to 29% for the whole region of Spain. As regards the two reduced regions presented in section 4.2 of the original manuscript, the DE reduces from 14.5% to 13% for the Spanish Plateau and from 25% to 22% for the Mediterranean Spanish coast at Valencia.

The reviewer also notes that the surprising high DE global value for Spain indicates that there are regions where the DE must be even higher. In this sense, the revised manuscript includes two new regions with usually high intensity storms. The subregions are indicated in the new modified figure 2 (shown below). The first region (with magenta color in the figure below) corresponds to a region including Canary Islands and the West African Atlantic coast covered by AEMET between [27° N, 37° N]x[20º W, 5° W], while the second one corresponds to the Alboran Sea, [35° N, 37° N]x[5º W, 0° W], at the South Mediterranean coast of Spain (with dark blue color in the figure). The first region includes a transition between the Atlantic Ocean, while the second one is a transition between land areas and a small sea, the Mediterranean Sea, including the Straits of Gibraltar, a region with frequent strong marine currents. The DE obtained for these regions is 49 % and 53 %, respectively, which justifies the high average new value for Spain of 29 %.

[Figure]

**Figure 2: Different areas for the studies presented in this work.**

According to figure 8 in the original paper showing the DE for different peak amplitudes, the DE value considerably increases with high energy strokes. In this sense, as kindly suggested by the reviewer and in order to try to explain the origin of the differences in the DE for the different subregions considered, the peak distribution of lightning strokes for each subregion has been calculated for this revised manuscript (similar to figure 7 but limited to the four subregions). A new figure with the results has been included (Fig. 10 shown below).

[Figure]

**Figure 10: Distribution of AEMET return strokes also detected by the WWLLN, in blue color, and total AEMET return strokes, in orange color for different subregions: a) Spanish Plateau, b) East Spanish Mediterranean coast, c) West African Atlantic coast, and d) South Spanish Mediterranean coast.**

It can be appreciated from them that the continental area at the Spanish Plateau presents an important distribution of lightning strokes at low energies, while the presence of high energy strokes increases in the other three areas containing land-sea transitions in the following order: East Spanish Mediterranean coast, West Atlantic region, and South Spanish Mediterranean coasts. This, combined with the results shown in Figure 8 in the original manuscript, seems to indicate that the DE is higher in land sea transitions influenced by a different energy distribution towards higher peak currents in the storms for those areas.

The revised manuscript includes the study for the two new subregions in section 4.2, the peak current distribution figure shown above, together with a discussion on the possible link between this energy distribution and the DE values.

**Reviewer's Specific Comments**

*Abstract First sentence is good, but the following text has too much detail for an abstract. Consider deleting sentences after the first down into line 22. Then resume with: This study finds the detection efficiency of WWLLN is around 38% . . . and continue with the remaining text in the abstract.*

**Response**

The paragraph has been reduced and reorganized to eliminate the excessive details but still providing a brief introduction of the WWLLN to researchers non-directly concerned with this global network.

*Tables*

*Caption on Table 1 declares a date range 2004–2022. However, datasets in the table are from 2003–2015, while publication references are from 2004–2018.*

*Table 1 has some historical interest, but could be shorter. Much of the contents is not relevant to WWLLN in 2012 or in the 2020s, because algorithms and network station distribution have changed greatly.*

**Response**

The caption has been corrected. As regards the reduction of contents, we think that the items included not only describe the historical evolution of the network performances, showing the differences in the working parameters and resulting DE for different studies, but also facilitates the understanding of the paragraphs describing the WWLLN feature evolution since its initial times (lines 160 and following in the original manuscript).

In this sense and pointing to the interest in including summarizing details in the table, Reviewer 1, in one of his comments, makes reference to a detail in the text that is not in the Table ( *L190: "...the best data recorded by WWLLN so far was a DE of 31%...". This value is not even in your Table 1*). It seems that summarizing capability of this table advises maintaining information even when it could be considered as non-relevant for highly specialized readers.

*Figures*

*Figure 2: Text near line 565 describes the cyan region of Figure 2 as the boundary for study, but Figures 10–12 show strokes outside the Figure 2 region. Is the text wrong? Or do Figures 10–12 show strokes not considered in the analysis?*

**Response.** The reviewer is right. The region only approximately corresponds to the cyan region of the study in section 4.2. It has been chosen to match the areas covered by the

maps provided by AEMET. In the revised version, the text has been changed accordingly. The figure numbers have been updated to Figs. 11 to 13 and, additionally, as suggested by Reviewer 1, the areas of the original figures 10b, 11b, and 11c have been adjusted to match the regions of AEMET shown in figures 11a, 12a, and 13a of the revised manuscript, respectively. The following sentence has been added to clarify this point at line 353:

> "The cyan region also approximately corresponds to the final monitoring application presented in section 4.3."

***Figure 5A:*** *horizontal axis label is wrong. West should be negative, but the label has west is positive.*

**Response.** Thank you for noting the mistake. The figure caption has been corrected.

***Figure 8:*** *shows a point above DE=1. Isn't that impossible? The method for calculating uncertainties cannot be correct, for an uncertainty bar extending above DE=1 is wrong. The smoothed blue line is difficult to see behind the red circles. If the blue line were plotted on top of the circles, both would be visible.*

**Response:**

As regards the point above DE=1, it was due to the mistake in calculating the DE the reviewer mentions in section 4, which provides an overestimate of the DE. This calculation has been corrected and the DE values are now below 1 as expected.

As regards uncertainties, the values above DE=1 result from direct statistical operations using the set of available data, which may lead to unphysical solutions. Of course, these statistical results must be completed with the condition that DE is lower or equal than unity. In Fig. 8 of the revised manuscript, the vertical axis has been redefined to avoid unphysical solutions and the blue line has been plotted on top of the circles to better appreciate it.

***Reviewer's Comments on Section 1, Introduction***

***Comment:*** *lines 69–75: The method for calculating DE seems to be incorrect (see comments for Section 4).*

*A CG stroke is not a vertical column above a point on Earth's surface. The path of a stroke often has a large horizontal displacement. Given this behavior of strokes, what is meant by stroke location? For a VLF stroke detection, this is an effective point for the transmitting antenna location. That point is unlikely to be the stroke contact point on the ground, and it is unlikely to be the effective location of an LF transmitting antenna. Stroke location is expected to be slightly different for different kinds of instruments and it is not defined or meaningful below some distance scale. These considerations mean that one must be careful in finding meaning in stroke accuracy, at small value, from different instruments.*

**Response.** As we mention above, the DE calculation has been corrected in the sense indicated by the reviewer's comments on section 4 and all related text, figures, tables…, have been modified accordingly.

As regards the care that must be taken when talking about location accuracy of a lightning stroke, a sentence clarifying the difficulty in defining the lightning location has been included in section 2 after line 117 in the original manuscript (124 in the revised one), where lightning location is described. The paragraph reads as follows:

> "… is simultaneously detected by a minimum of 5 stations. It is worth noting that care must be taken when interpreting data of lightning location below some length scale. This is so because the distance determined by WWLLN corresponds to an equivalent VLF antenna transmitting from an effective point, but the actual lightning stroke path is not usually a vertical one, thus, the distance detected will not exactly coincide with the stroke contact point on the ground. The meaning of this distance is even more approximate because of the comparison with independent AEMET results, obtained with Low Frequency (LF) technology, where characteristic distances differ from those of WWLLN."

**Reviewer's comments on Section 2, WWLLN**

*Comment: Lines 91–93, 119–120: Fig 2 of 10.1029/2020GL091366 (Lightning in the Arctic) shows the history of the count of active WWLLN stations. The number of active stations is the important number for network performance, and is always less than the number of stations. Some stations are offline at any time due to network, power, or other technical issues.*

**Response:** The sentence at line 91-93 in the original manuscript (now line 96) has been modified mentioning the active stations and a reference to (Holzworth et al., 2021) has also been added in the following terms:

> "The distribution of associated active stations around the whole Earth, slightly above active 60 stations since 2014 (Holzworth et al., 2021), makes it possible that the WWLLN achieve global location of lightning strokes at a planetary scale with a constantly improved accuracy…"

The sentence in line 119-120 (now line 133) has been modified in similar terms:

> "The number of active stations, the important number, increased until an almost stable number around 60 active stations since 2014, approximately

Other sentences concerning the number of active stations have also been slightly changed throughout the paper.

**Comment:** *Lines 120–126: For a recent comparison over time, WWLLN detection efficiency compared to New Zealand lightning network is shown in Fig 1 of 10.1029/2019JD030975 (Global Distribution of Superbolts).*

**Response:** The paragraph beginning in line 119 (now line 131) includes now a sentence with the recent comparison for New Zealand and the corresponding reference. The paragraph reads as follows:

> "The WWLLN had 40 receiving sensors in 2010, providing a DE of around 11 % in 2010 for peak currents greater than 20 kA (Abreu et al., 2010) and a LA of around 5 km. A recent comparison over time of the WWLLN detection efficiency for different peak currents can be found in (Holzworth et al., 2019) for the New Zealand area. The number of active stations, the important number, increased until an almost stable number around 60 active stations since 2014, approximately."

**Comment:** *Here and elsewhere (lines 416–418, 671–673), is the idea that a higher density of nearby stations might cause DE to be higher. Even assuming station density is high and that DE is high, there is no analysis in the paper showing a cause-effect relation. The link between WWLLN station density around Spain and the apparently higher DE is purely speculation in this paper, and that should be made clear wherever this possible link is mentioned.*

*Many stations nearby is offered as a possible explanation for the high DE around Spain. However, there are other considerations for stations near a storm that work against this explanation: (1) close strokes can saturate the receiver, distorting the waveforms; (2) close strokes have less frequency dispersion, making it harder to extract the time of group arrival; (3) when several nearby stations see a distant stroke, much of the information is redundant; (4) nearby strokes have more high frequency content that is noise in the VLF analysis—these high frequencies decay quickly with distance. In the stroke location algorithm, stations closer than 300 km to a stroke are not used.*

**Response:** The reviewer is right. The nearest stations are probably saturated when high energetic events happen (maybe this is not the case for the low intensity case, but the reviewer is right noting that the sentence was speculative). In addition, since we have no way of proving the link between the higher density of stations in Spain and the high results for DE, we will eliminate those unproved ideas from the paper.

In this sense, the following paragraphs have been modified or included:

Lines 72 and following:

> "Our work will show that the WWLLN provides values for the DE and LA in the area of Spain which are higher than those reported up to the moment, with remarkable results for high peak current lightning strokes. A subsequent second

study concerning four Spanish subregions with different geographical characteristics will be addressed to detect possible variations on the WWLLN performances and their link to differences in the energy distribution of lightning strokes at these areas."

Lines 139 and following:

"… world, the characteristics of the Spain, with important geographical differences in relatively short distances (coasts, islands, mountain ranges, an inland plateau region surrounded by mountain regions,...), may greatly affect the storm characteristics and, therefore, the WWLLN performances in relatively short distances"

The following sentence at the end of section 2, line 244, in the original manuscript has been removed (after line 281 in the revised manuscript)

"This high density may affect the features of the WWLLN in this region, not only because of an improved measurement capability, but also because of the increase in the available stations, which raises the chance of a stroke being simultaneously detected by at least five stations, and thus being registered as a valid stroke"

*Comment: lines 136–145: Some details in the list are wrong. No power or peak current estimates are produced in the WWLLN stroke analysis. Instead, stroke VLF (5–18 kHz) energy and uncertainties, in Joules, are output. Stations used for energy calculation are restricted to be in the range 1000–8000 km from the located stroke.*

**Response:** Dear reviewer, WWLLN provides information through files with different formats. We have used the APP files, which provide the information in the original manuscript. We have changed the sentence in line 151 of the revised manuscript to clarify that the information in the list corresponds to APP files.

"The WWLLN data are provided to customers and members of the WWLLN in different formats. The APP files used in this paper provide the following information for each lightning stroke detected:"

*Comment: Lines 157–158: Only stroke VLF energy is provided by WWLLN. An approximate linear relation between stroke VLF energy and peak current is sometimes used to estimate peak current. Of course, once a WWLLN stroke is matched to an AEMET stroke, peak current is available from the AEMET data.*

**Response:** Thank you for noting the error. The revised manuscript eliminates the sentence from original lines 157-158 and a new sentence has been added at line 495 of the revised manuscript to clarify that the peak current of the WWLLN strokes is that corresponding to the matched AEMET strokes. The paragraph reads as follows:

"… of the detected CG strokes. As regards the WWLLN, the peak current is assigned as the one corresponding to the matched AEMET stroke. The same ratio between positive …."

***Comment:*** *lines 167–168: (DE and LA) must be considered as global value, since they correspond to the detected lightning strokes, independently of their current peak amplitude. If I understand what this sentence should mean, global is a confusing word to use here. Possible rewording: (DE and LA) values were calculated from all the matched lightning strokes, independently of their peak current.*

**Response:** This comment is also made by the first reviewer. We agree that WWLLN is a global network and the term global for DE may not be appropriate. We have changed "global" by "total detection values", and slightly modified the sentence after line 179 of the revised version in the following terms:

"It is worth noting that these two quantities must be considered as total detection values, i.e., they correspond to all the detected lightning strokes, independently of their current peak amplitude. More detailed information on the DE values for specific current peak amplitudes can be found in the works referenced in Table 1"

The term "global" has also been changed to "total" in the revised line 228 for the same reason.

***Comment:*** *line 185: DE in the early days of WWLLN was about 1%, but DE at these early times in the development of the network are not relevant for this study (2012 and 2020–2022). For 2012, WWLLN global DE was probably above 10%. For 2012, the median number of active stations on any day was 55. The total number of stations, 69, is a less useful number.*

**Response:** We think that the historical review of WWLLN state of development and performances may be useful for interested readers, but the reviewer is right in noting that this low value corresponds to the early stage of WWLLN and that it soon improved the value above 10% around 2012. The sentence has been modified in the following sense (line 198 in the revised version):

"Focusing on the results shown in Table 1 and references therein, they report an initial very low DE for the early WWLLN measurements in 2003, which was in the order of the one percent of the total lightning strokes detected by the reference networks, and reached values around 10 % for year 2012, the object of this study."

**Reviewer's comments on Section 3, AEMET**

***Comment:*** *line 294: Are AEMET stroke times rounded to the nearest second or truncated to the second? For instance, if a stroke was originally time-stamped at 01:23:45.678901 at*

*microsecond accuracy, is that stroke's AEMET time published as 01:23:45 or 01:23:46? The method used affects DE.*

**Response:** We have no information on that aspect.

*Comment: lines 299–301: The description for how AEMET IC and CG strokes were separated is incomplete. How exactly was this done? How sensitive are DE calculations to the criteria used to separate strokes types?*

**Response:** Dear reviewer, in lines 301-311 of the original manuscript (now 329 to 338), we give references where this issue is addressed. The reader can go further reading these references. We assume that AEMET data is CG. This was explained also to reviewer 1. The WWLLN data include both IC and CG. From our knowledge, there is a ratio IC/CG that is obtained doing special surveys.

**Reviewer's comments on Section 4, WWLLN performance in Spain**

*Comment: lines 406–411: The calculation of WWLLN DE relative to AEMET appears to be incorrect, in a way that over-estimates DE. The correct DE calculation is to count the number of AEMET strokes that match a WWLLN stroke, then divide by the total number of AEMET strokes. This ratio is the fraction of AEMET strokes also detected by WWLLN—a detection efficiency. The method in the manuscript finds the number of WWLLN strokes that match an AEMET stroke, then divides that by the total number of AEMET strokes. It is a ratio of WWLLN strokes to AEMET strokes, and is not a detection efficiency. It over-estimates DE because it is possible for several WWLLN strokes to match with one AEMET stroke, especially within a 1 second matching time window. Because WWLLN cannot detect the same AEMET stroke more than once, the numerator of the ratio becomes larger than it should be, and DE is over-estimated.*

**Response:** Thank you for noting this error. As we mention in our previous response to your general comments, the DE has been recalculated in the correct way you indicate, i.e., by finding which AEMET strokes were also detected by WWLLN.

The following table shows the result for the DE obtained for Spain and the four subregions considered: the Spanish Plateau, the West Mediterranean coast, and the two new subregions considered, namely, the South Mediterranean coasts and West Atlantic region. The table shows the values obtained with the original method (incorrect) and the corrected one.

**Some details of Table 2 for the revised manuscript**

| Region | Erroneous DE (%)) | Corrected DE (%) |
|---|---|---|
| Spain (orange in Fig.2) | 38 | 29 |
| Spanish plateau (green in Fig. 2) | 14.5 | 13 |
| East Spanish Mediterranean coast (cyan in Fig. 2) | 25 | 22 |
| West African Atlantic coast (magenta in Fig. 2) | 67 | 49 |
| South Spanish Mediterranean coast (dark blue in Fig. 2) | 79 | 53 |

Of course, the revised manuscript only includes the correct values of the DE for the five regions in a revised Table 2 and throughout the text. Details of the location accuracy and confidence interval along the West-East and South-North directions are also included in the modified Table 2.

*Comment: lines 420–423: Although there is a resemblance, this is not a Rayleigh distribution. The tail is too heavy; Rayleigh falls fast, as exp(−x2 ). This distribution is also too narrow around its median value. Graphing a few Rayleigh distributions and comparing with the histogram shows this easily. Statistical methods, such as χ2 goodness-of-fit, would confirm this numerically.*

**Response.** The resemblance is only qualitative and does not introduce valuable information, therefore the following sentence has been removed:

> "It resembles a clear Rayleigh distribution, a distribution typical for nonnegative-valued random variables. This distribution is often observed when the over-all magnitude values are related to two independent components. This is our case, where the location error depends on two parameters, latitude and longitude"

*Comment: lines 549–553: The claim of DE/LA differences in the two geographic regions seems plausible, but there is no evidence or a clear line of reasoning that explains how geography relates to DE/LA differences. For example, a study of the WWLLN stroke energies or the AEMET peak currents in the two regions could explore whether different*

*stroke energy distributions explain the DE differences. The assertion that there are more intense atmospheric phenomena occurring in this region should have a reference. There is only speculation here, with the goal stated in the abstract, lines 25–26, not being met.*

**Response:** The reviewer is right in noting the speculative nature of our statement. It is based on our particular knowledge of differences in storms in Spain, the origin of the authors, but this is not a proper reference to support such a statement. The reviewer wisely proposes a study of stroke energy distribution for each region. As we mention before in this response, when answering the general reviewer's comments, we have done this study. A new figure 10 has been included and discussed. It can be seen from this figure a difference in the energy distribution: regions with more energetic strokes correspond to regions with higher DE values. The new figure and the possible explanation for the differences in DE values have been included in the revised manuscript. The text in the paper has also been modified accordingly, mainly in section 4.2, but also in the introductory and conclusion sections.

**Section 4 Conclusions**

*Comment: lines 662–663: Number of stations in 2012 compared with now: what matters is number of active stations, not number of stations. Both have grown over the years. A correct statement is that based on WWLLN growth, it is reasonable to assume that the 2012 DE is a lower bound to the present DE in Spain. However, if the AEMET network has also improved enough, then WWLLN DE relative to AEMET might even be lower now than in 2012.*

**Response:** Thank you for the comment, you are right. The first paragraph in the conclusions section, after line 713, has been modified in the following terms:

> "The current number and distribution of the WWLLN stations, around 70 stations with around 60 active ones, is similar to that considered in the study with data from 2012, therefore, results presented here are valid nowadays although, based on WWLLN growth, it is reasonable to assume that the 2012 DE is a lower bound to the present DE in Spain. Moreover, if the evolution of the AEMET network has surpassed the evolution of the WWLLN, the DE relative to AEMET might even be lower now than in 2012."

Additionally, the first paragraph after line 338 in the original manuscript has also been changed to clarify that the study of 2012 must be understood as a lower bound to present performance of the WWLLN. The paragraph includes now the following sentences starting at line 360 of the revised manuscript:

> ".... First, data from AEMET were openly available to the authors at that period, and second, this period was close in time to the moment at which the station in

Valencia (Spain) was deployed by the team of this work in the year 2011. It must be noted that the objective of this paper is not comparing AEMET and WWLLN in order to determine which is the best network, since they have different local and global objectives, respectively. The main goal is to determine the DE of the WWLLN in the whole region of Spain, which includes continental and insular regions, using the AEMET data as true data (Abarca et al., 2010). This study is the first one analyzing the WWLLN performances at Western Europe, specifically at the Spain, where large geographical differences happen at relatively low distances. The use of data from 2012 does not invalidate the conclusions of the first study, although, bearing in mind the technical improvements of both WWLLN and AEMET since 2012, the results presented here must be considered as a lower bound for the current network performances, which, most likely, will have been improved at present days "

**Comment:** *line 688: How does one decide that there is good agreement between AEMET and WWLLN? This seems like a subjective evaluation, which is ok. But it should be clear this is an opinion rather than a rigorous finding*

**Response:** The reviewer is right, we have changed the sentences to clarify that this is a qualitative comparison based on visual comparison, not an objective quantitative one. Concretely, in section 4.3, the paragraph beginning in line 582 of the original manuscript has been modified as follows (line 693 of the revised manuscript)

"In our opinion, an acceptable qualitative match is observed, although it must be noted that a rigorous statement on the quality of the results would require a quantitative comparison more than mere image comparisons such as those shown in Figs. 11 to 13. Bearing this subjective and approximate sense in mind, a good reasonably concordance can be appreciated …",

and the conclusions section has been modified as follows (line 748 of the revised manuscript):

"The study of three severe storms which affected the Mediterranean Spanish coast at Valencia during years 2020, 2021, and 2022 seems to show a qualitative good agreement with screenshot results available from the AEMET national agency used as reference in this work"

The typos/grammar have been corrected. Thank you for letting us know.

Other minor grammar changes have also been corrected to clarify some minor points and improve the readability of the paper.

---

## Referee Report (RR1)

Review of revised manuscript 10.5194/egusphere-2024-704

**The World Wide Lightning Location Network (WWLLN) over Spain**

Reviewer comments and questions were thoughtfully addressed, and appropriate changes to the methodology and text were made.

There is an apparent arithmetic error or typo, without consequence to the results of the paper: lines 412–413 describe 30MB files for the time 01Jan to 30Apr (say 120 days), implying 3.6GB of data, yet the text refers to a much larger 8.1GB of data.

Some additional information on APP files, again not affecting the results of the paper, but perhaps useful for future work:

1. it looks as if only 123 APP files were ever produced, all during the time period 01Jan–06May2012.

2. APP power is calculated by adding up stroke energy and then dividing by an integration time. As such it is not a peak power, but a 1.3ms-average power. It was recognized early on that this averaged power is less physically meaningful than stroke energy.

3. APP files have been superseded by AE files, which are explicitly a stroke energy product, and have been produced from 15Apr2009 to the present.

---

## Author Response (AR2)

**Dear Editor:**

Thank you for your kind invitation to submit a minor changes revised version of our manuscript for its publication in Natural Hazards and Earth System Sciences.

**Manuscript Details:**

**egusphere-2024-704**

**Title: The World Wide Lightning Location Network (WWLLN) over Spain**

**Author(s): Enrique A. Navarro et al.**

**MS type: Research article**

First of all, the authors would like to thank the reviewers for their valuable comments, which have helped to improve the quality of the manuscript.

**A minor change has been made attending reviewer comments referee-report-2:**

- There was a typing error in line 413, the value 8.1Gb was changed to 3.96 Gb in the revised manuscript.

We also will take all comments into account for future work.

Yours sincerely.

Enrique A. Navarro

---

## Author Response (AR3)

**Dear Editor:**

Thank you for your kind invitation to submit a technically corrected version of our manuscript for its publication in Natural Hazards and Earth System Sciences.

> **Manuscript Details:**
>
> **egusphere-2024-704**
>
> **Title: The World Wide Lightning Location Network (WWLLN) over Spain**
>
> **Author(s): Enrique A. Navarro et al.**
>
> **MS type: Research article**

**A technical correction has been made attending Editor's comment:**

Figures 11, 12, 13, were not entirely created by the authors, therefore captions of these Figures are corrected to include credit.

Yours sincerely.

Enrique A. Navarro